

# Review article: Stocktaking of methods for assessing dynamic vulnerability in the context of flood hazard research

Julius Schlumberger[1,2], Tristian Stolte[2], Helena M. Garcia[3], Antonia Sebastian[3,4], Wiebke Jäger[2], Philip Ward[1,2], Marleen C. de Ruiter[2], Robert Šakić Trogrlić[5], Annegien Tijssen[1], and Mariana Madruga de Brito[6]

[1]Climate Adaptation and Disaster Risk Department, Deltares, The Netherlands
[2]Institute for Environmental Studies, Department of Water & Climate Risk, Vrije Universiteit Amsterdam, The Netherlands
[3]Environment, Ecology and Energy Program, University of North Carolina at Chapel Hill, USA
[4]Department of Earth, Marine and Environmental Sciences, University of North Carolina at Chapel Hill, NC USA
[5]International Institute for Applied Systems Analysis (IIASA), 2361 Laxenburg, Austria
[6]Department of Urban and Environmental Sociology, UFZ-Helmholtz Centre for Environmental Research, Leipzig, Germany

**Correspondence:** Julius Schlumberger (julius.schlumberger@deltares.nl)

**Abstract.** Dynamic vulnerability, driven by changing social, economic, physical, and environmental characteristics, is critical to understanding flood risk. Despite its importance, existing flood risk assessment research often overlooks the mechanisms that drive dynamic vulnerability and the interactions between underlying characteristics. In this study, we systematically review methods used to assess dynamic vulnerability in the context of floods and compile their findings about the drivers and effects of the dynamics in a dataset. We identify 28 relevant studies and group them into four categories of vulnerability dynamics:
single-event, consecutive events, co-occurring events, and underlying dynamics. We find that most studies rely on indicator-based, statistical, or qualitative methods, with a notable under-representation of damage curves and process-based modeling approaches such as agent-based models. Demographics, economic characteristics, and awareness of flood risks are vulnerability dimensions most frequently assessed, whereas governance, health, crime, and conflict are rarely addressed. Data sources vary widely, with interviews and surveys dominating studies on consecutive events and single-event dynamics. In contrast, studies on underlying dynamics and co-occurring event dynamics use a much wider array of data sources (e.g., cadastral data, maps, or modeled data). This review highlights methodological gaps, including the limited analysis of causal relationships and the lack of integrated approaches for multi-hazard contexts. Advancing flood risk research requires holistic assessments, integration of diverse dimensions, and the development of dynamic modeling techniques to capture evolving vulnerability processes.

## 1 Introduction

Floods rank among the most significant natural hazards globally in terms of their impacts (IFRC, 2023; Rentschler et al., 2022). Since 2000, the number of flood events reported in the CRED EM-DAT database has more than doubled (WMO, 2021). This increase is partly attributed to growing exposure to flood-prone areas because of population increase and urbanization (Rentschler et al., 2022). It also might be influenced by the effects of climate change (Hirabayashi et al., 2021). Notably, the population in flood-prone regions grew by about 20-24% between 2000 and 2015 (Tellman et al., 2021). At the same time,



progress in reducing flood vulnerability has not been sufficient to counterbalance the increased exposure. Still, it can potentially minimize flood impacts, particularly with robust protective infrastructure (Sauer et al., 2024).

Understanding and tackling vulnerability is crucial to mitigating the catastrophic effects of floods. Vulnerability is a multifaceted concept with varying interpretations. For example, some studies, such as those using a Coastal Vulnerability Index (e.g., Mclaughlin and Cooper, 2010), frame vulnerability in terms of physical susceptibility to hazards, often emphasizing geomorphological and hazard-related factors while disregarding socioeconomic dimensions. In contrast, broader frameworks, including those by the UNDRR (2017) and IPCC (2022), conceptualize vulnerability as a distinct dimension of risk alongside hazard and exposure, incorporating social, economic, and institutional factors. For this study, we adopt the definitions by the UNDRR and IPCC, as they provide a more comprehensive understanding of vulnerability in the context of flood risk assessment. Under this risk framework, vulnerability refers to the social, economic, and physical characteristics of an element at risk that make it susceptible to harm in the event of exposure to a hazard (IPCC, 2022). Flood vulnerability analyses commonly focus on quantifying the susceptibility of infrastructure and buildings to damage or isolating the sociodemographic or economic factors that influence human health and well-being in the event of a disaster (Merz et al., 2010). However, vulnerability can also evolve due to changing demographics, varying socioeconomic conditions, or experiences with and recovery from past impacts (Alwang et al., 2001). As a result, vulnerability is multi-dimensional, dynamic, and highly context-dependent (Cutter, 1996).

The dynamic nature of vulnerability has prompted calls for studies that account for its spatiotemporal evolution (Handmer et al., 1999; Simpson et al., 2021; de Ruiter and van Loon, 2022; Stolte et al., 2024). While various methods for assessing vulnerability exist (for an overview, see e.g. Douglas, 2007; de Ruiter et al., 2017; Hagenlocher et al., 2019), recent studies highlight significant gaps in capturing its dynamics (Moreira et al., 2021; de Ruiter and van Loon, 2022; Jurgilevich et al., 2017). These challenges are evident in assessments of single-hazard vulnerability (e.g., vulnerability to floods) and in understanding the interactions of vulnerabilities in multi-hazard contexts, such as triggering, amplification, or cascading hazards (Gill et al., 2022; Schlumberger et al., 2024; Šakić Trogrlić et al., 2024).

Given the growing emphasis on understanding flood vulnerability and its underlying drivers, reviewing advancements in flood risk research, including the methods employed and available data, is essential. Previous reviews have partially explored the temporal vulnerability dynamics but primarily focus on the "what" of assessments, leaving critical questions about the "why" and "how" of its evolution unanswered. For instance, Moreira et al. (2021) categorized flood vulnerability indices into pre-event, event, and post-event phases, highlighting a predominant focus on pre-event vulnerability with limited attention to post-event assessments (e.g. Carlier et al., 2018; Miguez and Veról, 2017). Similarly, Drakes and Tate (2022) systematically reviewed which subdimensions of social vulnerability have been considered outcomes versus those assumed to be preconditioned in consecutive, co-occurring, or aggravating multi-hazard scenarios.

To address these gaps, this study provides a comprehensive overview of approaches for assessing dynamic flood vulnerability. We identify case studies that explicitly assess vulnerability as a dynamic process, examining the methods and data sources used. Additionally, we analyze which (sub)dimensions of vulnerability are incorporated and identify patterns and gaps





in current practices. Our study offers a roadmap for advancing more robust and dynamic flood vulnerability assessments by synthesizing existing approaches and highlighting critical gaps.

## 2   Methods

### 2.1   Applied Concepts and Scope for the Analysis

We identify four categories of dynamic vulnerability essential for flood risk assessment (Figure 1). We use them to investigate
whether specific methods or data are more prevalent in studies aiming to address one of the vulnerability categories than another. Our definitions build on those identified by de Ruiter and van Loon (2022), who categorized vulnerability dynamics into (a) vulnerability dynamics from underlying (non-hazard specific) processes, (b) vulnerability dynamics from long-lasting disasters, and (c) vulnerability dynamics from compound or consecutive events. We further refine these categories, and add a fourth to capture better the mechanisms influencing dynamic flood vulnerability:

– **Single-event dynamics:** Changes in vulnerability in response to a single flood event, such as physical damage to buildings or injuries that reduce capacity to future stresses. For example, Thomson et al. (2023) demonstrated how financial vulnerability can change after a flood event by simulating mortgage default risks following Hurricane Florence, where uninsured losses and property devaluation increased the likelihood of abandonment and financial instability for affected homeowners.

– **Consecutive-event dynamics:** Changes in vulnerability due to the overlapping effects of consecutive flood events and ongoing recovery processes. For instance, a partially damaged building may respond differently to subsequent flood events (i.e., physical vulnerability) or individuals with prior flood experience may react differently to warnings (i.e., social vulnerability).

   – **Co-occurring event dynamics:** Changes in vulnerability due to simultaneous hazards, at least one of which is a flood.
For example, a farmer experiencing both flooding and pandemic lockdowns may face compounded vulnerabilities due to limitations in finding field workers and in experiencing damages to their equipment and vegetables (Begum et al., 2023)

   – **Underlying dynamics:** Changes in vulnerability due to non-hazard specific or long-term factors such as socioeconomic development or conflict. For example, financial vulnerability may increase during economic crises, reducing an individual or community's capacity to invest in adaptation measures (Matanó et al., 2022).

### 2.2   Review process

Figure 2 highlights the process for conducting a systematic literature review of dynamic flood vulnerability. While systematic, the review did not strictly adhere to specific protocols, similar to a semi-systematic literature review as defined by Snyder (2019) or a meta-narrative review as defined by Wong et al. (2013). This allowed more flexibility in dealing with the emerging



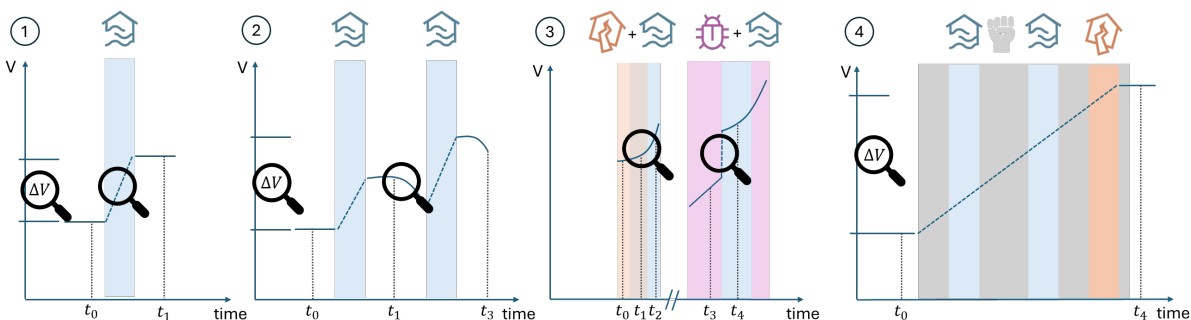

**Figure 1.** Vulnerability dynamics considered in this review. Single-event dynamics (focuses on changes in vulnerability before, during, and after a single flood event), consecutive-event dynamics (assesses recovery or combined effects of partial recovery and a new event), co-occurring event dynamics (examines how multiple simultaneous hazard-related drivers combine and interact influencing vulnerability), and underlying dynamics (analyses long-term changes in vulnerability without disentangling individual event-related dynamics). In this Figure, V = vulnerability; $\Delta V$ = change in vulnerability; $t_i$ = moment in time. Note that changes in vulnerability can be positive or negative. A magnifying glass denotes the most characteristic part of each vulnerability dynamics category.

field of dynamic vulnerability, where definitions and concepts vary widely within and across research communities. A search
query in Google Scholar on November 13, 2024, used keywords related to dynamic vulnerability and multi-hazard vulnerability assessment (Table 1). We also invited collaborators to suggest additional studies. This process yielded 980 publications.

**Table 1.** Overview of the applied search terms on Google Scholar and yielded results.

|  | N of publications |
|---|---|
| "dynamic vulnerability assessment" AND (flood OR floods OR flooding OR "flood event" OR "flood events" OR "floods") | 100 |
| "multi-hazard vulnerability assessment" AND (flood OR floods OR flooding OR "flood event" OR "flood events" OR "floods") | 315 |
| "multi-hazard vulnerability analysis" AND (flood OR floods OR flooding OR "flood event" OR "flood events" OR "floods") | 19 |
| "dynamic vulnerability analysis" AND (flood OR floods OR flooding OR "flood event" OR "flood events" OR "floods") | 85 |
| "vulnerability dynamics" AND (flood OR floods OR flooding OR "flood event" OR "flood events" OR "floods") | 419 |
| Additional papers added by collaborators | 41 |
| **Total** | **980** |

To be included, publications had to meet the following criteria: (i) published in English; (ii) peer-reviewed; (iii) freely accessible to the reviewers; (iv) investigate vulnerability concerning floods, potentially amongst other hazards; (v) adopt a





definition of vulnerability consistent with the IPCC (2022) or UNDRR (2017); (vi) addressing one of the vulnerability dynamics
identified in Figure 1; (vii) provide details on vulnerability assessment processes, data, equations, or methodologies, allowing
replication; and (viii) is a case study. As a consequence of criteria (v) and (vi), we exclude research that interprets changes
in exposure as part of vulnerability, such as studies examining changes in risk due to population growth (e.g., Ballesteros and
Esteves, 2021; Herslund et al., 2016; Ku et al., 2021; Londe et al., 2015) or hazard exposure due to coastal erosion/sea level rise
(e.g., Hastuti et al., 2022; Hoque et al., 2019; Islam et al., 2020; Kantamaneni et al., 2018). This narrower definition enhances
the comparability of the included studies (UNDRR, 2017).

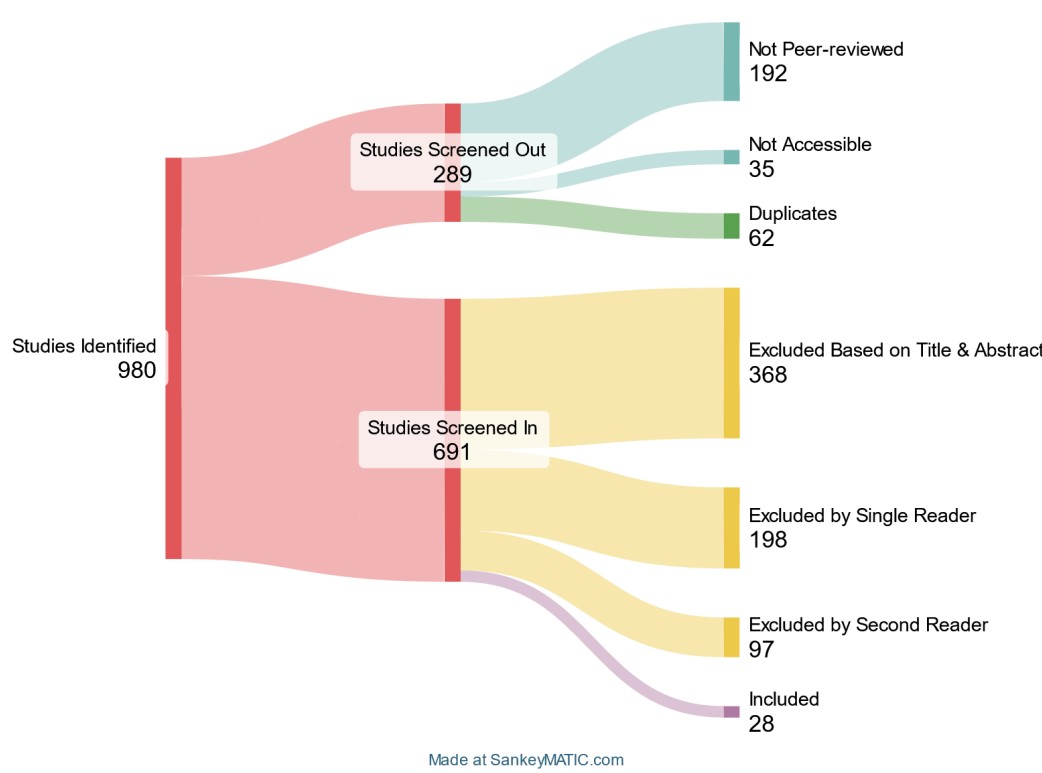

**Figure 2.** Summary of the literature review and number of publications included in this analysis.

Following the study selection, we performed a full-text analysis using the classification system in the Appendix in Table A1.
This analysis examined the assessment methods, elements-at-risk, and data used in each study. We categorized the methods
into five groups, extending a framework by Nasiri et al. (2016):

1. Indicator-based methods aggregate data into vulnerability indicators (e.g., de Brito et al., 2017; Kappes et al., 2012;
Moreira et al., 2021).





2. Curve methods relate hazard intensity (e.g., inundation depth) to damage inflicted (e.g., Arrighi et al., 2020; Fuchs et al., 2019; Tarbotton et al., 2015).

3. Process-based modeling methods use system or process-based approaches to capture causal relationships between hazards, vulnerability, and impacts (e.g., Lu et al., 2023; Dzulkarnain et al., 2019; Joakim et al., 2016).

4. Disaster impact data methods leverage historical flood impact data to estimate vulnerability (e.g., Mechler and Bouwer, 2015; Tanoue et al., 2016).

We first removed duplicates, inaccessible publications, and non-English or non-peer-reviewed works to identify relevant papers meeting these criteria. Next, we excluded irrelevant studies based on title and abstract. Finally, at least two authors reviewed each of the remaining publications for relevance, focusing on the categories of dynamic vulnerability defined in Section 2.1. Decisions on their relevance were made collectively based on the inclusion criteria. Through this double-review process, 28 papers were identified as relevant, applying some form of dynamic vulnerability assessment in a specific case study. Due to the relatively small number of studies in each category, conducting meta-analyses or statistical comparisons was not feasible. Instead, we provide a qualitative description of key studies and their methodological approaches.

Statistical analysis methods analyze correlations between data and vulnerability dynamics. Qualitative analysis methods use narratives and expert knowledge to describe cause-effect relationships qualitatively (de Ruiter and van Loon, 2022; de Brito et al., 2024). To address varying definitions of vulnerability, we characterize the vulnerability by the dimensions considered in the study, focusing on the physical and social dimensions, as recently applied by Stolte et al. (2024). The physical dimension of vulnerability refers to the physical properties of elements at risk (de Ruiter et al., 2017), whereas the social dimension refers to the characteristics of a person or group in terms of their capacity to anticipate, cope with, resist, and recover from the impact of a flood (Wisner et al., 2004). Each dimension is, in turn, divided into several subdimensions of vulnerability.

## 3   Results

We identify 28 relevant studies that assess dynamic vulnerability. Most studies address underlying vulnerability dynamics, followed by consecutive and co-occurring dynamics as shown in Figure 3a. Single-event dynamics are covered the least often. The temporal distribution of studies shows an exponential growth in the number of publications that meet our criteria over the past four decades, especially since 2010 (Figure 3b). The earliest study dates to 1988 (Phifer et al., 1988) and examined consecutive event dynamics, but no additional studies meeting our criteria were published until 2010. After that, the number of publications grew exponentially and roughly quadrupled between 2015 and 2024.

As shown in Figure 3c, most studies (n=10) focus on European contexts, particularly Germany. Several other studies adopt a global scope, but their implementations vary widely. For instance, Formetta and Feyen (2019) and Jongman et al. (2012) utilize global datasets or models, while Kreibich et al. (2017, 2023) analyze case studies from various countries.

Figure 4 summarizes the general approaches, methods, dimensions of analysis, and data sources used. The following subsections provide detailed insights into these studies, grouped by the category of vulnerability dynamics they address.



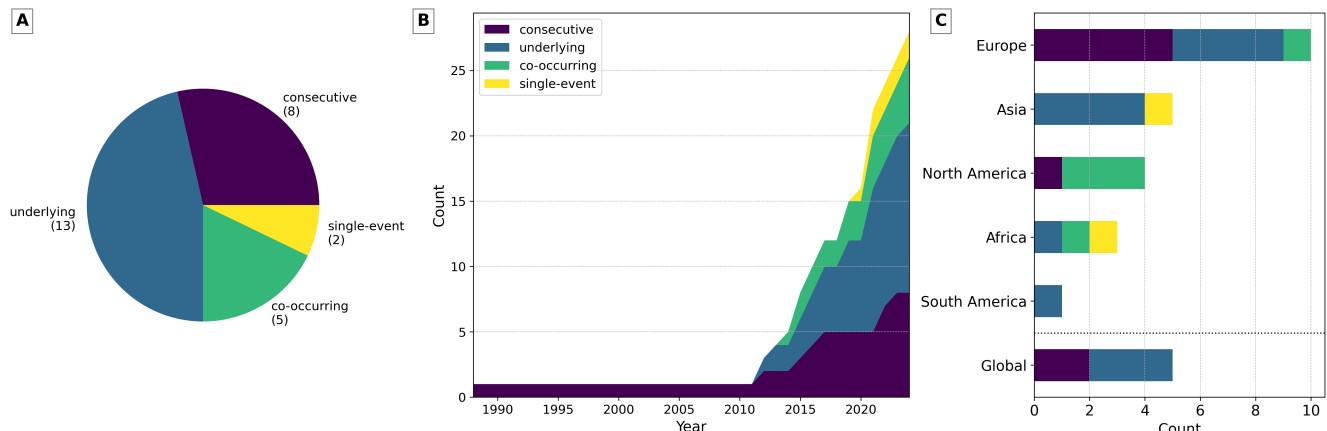

**Figure 3.** Distribution across categories (A), the temporal evolution (B), and the geographic distribution (C) of studies that develop or apply methods for assessing dynamic vulnerability. The colors indicate different categories of dynamic vulnerability assessed in each case study.

## 3.1 Single-event dynamics

Two studies assess vulnerability dynamics due to single flood events (Jamshed et al., 2021; Salvucci and Santos, 2020). The studies account for different elements of the social dimension of vulnerability to examine how flood events drive changes in vulnerability using surveys and statistical methods (see Figure 4). Jamshed et al. (2021) conduct retrospective surveys with 384 households in Pakistan and apply regression models to determine factors affecting rural-urban linkages and poverty as drivers of vulnerability. Likewise, Salvucci and Santos (2020) used a four-wave national household panel survey from 2014-2015 with 11,600 households to investigate the impact of the 2015 Mozambique flood on household consumption and poverty levels. They use the causal inference methods employing a difference-in-difference approach (Angrist and Pischke, 2009) to quantify changes in vulnerability attributable to the flood event.

Both studies focus on human life and health as the primary element at risk. Jamshed et al. (2021) investigate how flooding, directly and indirectly, impacts dependence between rural and urban communities and how this affects the flow of finance, information, goods, and people. Similarly, Salvucci and Santos (2020) observe that consumption shortly after flood events decreases significantly, especially for poorer households, increasing their vulnerability to future hazards.

## 3.2 Consecutive event dynamics

Eight studies assess the vulnerability dynamics due to consecutive events, including the processes that influence recovery. Most studies (n=4) rely solely on survey data, while the others consider solely literature or reports (n=2) or a combination of survey data and literature (n=1) or survey data and modeled data (n=1).

The studies vary widely regarding the time intervals between the consecutive events, the number of flood events considered, and the duration of the analyses (Figure 5). Most focus on recent floods, with only a few examining multiple decades (e.g.,





**Figure 4.** Summary of key characteristics of dynamic vulnerability assessment studies (n=28), including the category of vulnerability dynamics covered, the general approach followed, the physical and social dimensions considered, and the type of data used. The heatmap shows the frequency of overlaps between different characteristics of studies where each subplot represents a specific pair of characteristics (e.g., "Dynamics" vs. "Method").

Kreibich et al., 2017, 2023; Bubeck et al., 2012) and one covering the past century (Schoppa et al., 2024). The period between consecutive events (solely flood-flood consecutive multi-hazards were found) ranges from one to 42 years (average: 9.7 years,





median: 5 years). The timing of data collection for the assessment ranges from a few months to 29 years after a flood (average:
8.3 years, median: 7 years). It is important to note that Kreibich et al. (2017, 2023) use a literature review considering various
reports and publications from different years to assess vulnerability dynamics, while we took the date of the scientific publica-
tion to determine the time lag between events and data collection. If excluding these two studies, the timing of data collection
for the assessment ranges from a few months to seven years after the last flood (average: 1,8 years; median: 1 year). Further-
more, it is worth mentioning that some studies capture multiple flood events without collecting data between all consecutive
events (e.g., Köhler et al., 2023; Kienzler et al., 2015). Only one study considers pre-event data and collects data at multiple
moments between consecutive flood events (Phifer et al., 1988).

Four studies use statistical methods to investigate vulnerability dynamics. Phifer et al. (1988) apply factor and hierarchical
regression analysis on data from 200 older people in south-eastern Kentucky, USA, to investigate factors influencing health
changes at different timings after consecutive flood events. Köhler et al. (2023) apply linear and logistic regressions on sur-
vey data from 2462 residents in Saxony, Germany, to explore relationships between flood experience, adaptive behavior, and
self-reported resilience. Similarly, Kienzler et al. (2015) and Bubeck et al. (2012) use descriptive statistics (mean, frequency
distribution) on multi-wave survey data (between n=461 and n=1697) and 752 computer-aided telephone interviews. Three
studies rely on literature and reports (e.g., Thieken et al., 2016; Kreibich et al., 2017, 2023). Thieken et al. (2016) use flood-
related parliamentary inquiries, policy documents, and laws, reports of relief and aid organizations, expert workshops, and a
survey (n=1652 private households and n=557 companies) to investigate how changes in flood risk management between 2002
and 2013 influenced the outcome of the 2013 German floods. Kreibich et al. (2017, 2023) apply a meta-analysis approach, com-
paring reports and studies regarding single flood events to identify changes in vulnerability related to preparedness, awareness,
and crisis management. Finally, Schoppa et al. (2024) develop a process-based flood risk model. They integrate household loss
data with telephone survey responses on awareness and preparedness (n=597) using Bayesian Inference to create continuous
data across years with and without data to model flood risk changes over 120 years in Dresden, Germany. The authors also
develop a process-based socio-hydrological model using system dynamics and differential equations to capture the temporal
dynamics due to consecutive flood events.

The studies mentioned above capture the vulnerability dynamics regarding different elements at risk and vulnerability di-
mensions. While Phifer et al. (1988) focus on human health and well-being, taking into account demographic, economic, and
health dimensions of vulnerability,

### 3.3 Co-occuring event dynamics

Five studies assess vulnerability dynamics due to floods co-occurring with other hazard types, including windstorms (Sarker
and Adnan, 2023; Bernier and Padgett, 2019), hurricanes (van Verseveld et al., 2015), the COVID-pandemic (Whytlaw et al.,
2021; Albulescu and Armaș, 2024), and droughts (Bola Bosongo et al., 2014).

Input data vary significantly across studies. Some rely on expert knowledge to qualitatively analyze changing vulnerabilities
during co-occurring events (Whytlaw et al., 2021; Albulescu and Armaș, 2024) or use it as input for Fuzzy Analytic Hier-



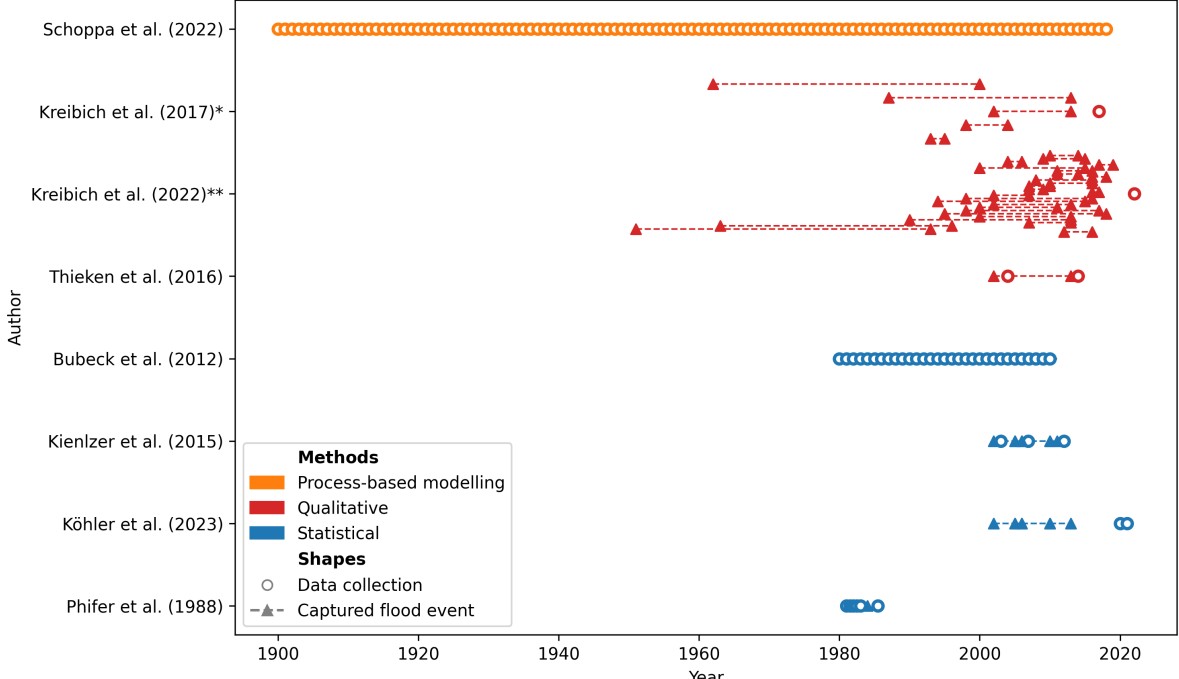

**Figure 5.** Timing of consecutive flood events (triangles) and the timing of data collection (circles) for different studies grouped by color according to the methods used. *Kreibich et al. (2017) reports on five separate consecutive events. ** Kreibich et al. (2023) analyses 26 separate consecutive flood events. Schoppa et al. (2024) uses a process model that assesses vulnerability on a yearly timestep.

archy Process (Sarker and Adnan, 2023). These studies combine expert knowledge with secondary data to link hazards with vulnerabilities and impacts. Bola Bosongo et al. (2014) use primary data from interviews and surveys (n=144) and secondary data from public administration to compare impacts during years with co-occurring floods and droughts against years with only droughts. As such, they offer a comparative view of vulnerability during multi-hazard events using vulnerability indicators. Bernier and Padgett (2019) investigate the relationship between storage tanks' response to waves, wind, and floods using mechanistic models and modeled data to determine tank failure due to different drivers. van Verseveld et al. (2015) establish relationships between observed damages and multiple hazard indicators using Discrete Bayesian Networks to develop predictive impact models.

The studies target a wide range of elements at risk. Whytlaw et al. (2021) and Sarker and Adnan (2023) focus on human life and well-being in the context of evacuation in the US and Bangladesh, respectively. Bola Bosongo et al. (2014) assess the socioeconomic impacts of floods on farming in Zimbabwe, while van Verseveld et al. (2015) examine housing building damages in New York City, US, and Bernier and Padgett (2019) assess the impacts on storage tanks in the context of petrochemical industrial facilities in Texas, US. Albulescu and Armaș (2024) use augmented impact chains to express the effects of hazard impacts and risk mitigation measures on vulnerability without focusing on a certain element at risk.





### 3.4 Underlying dynamics

Thirteen studies explore changes in vulnerability over time due to underlying (i.e., non-hazard triggered) socioeconomic dynamics (Meijer et al., 2023; Cian et al., 2021; Jalal et al., 2021; Jurgilevich et al., 2021; Rahaman and Esraz-Ul-Zannat, 2021; 205 Formetta and Feyen, 2019; Fekete, 2019; Araya-Muñoz et al., 2017; Jongman et al., 2012; Tanoue et al., 2016; Giupponi et al., 2013; Menoni et al., 2012; Li, 2024). The temporal extents of these studies (Figure 6) range from nine to 70 years, with resolutions between one year and two discrete moments in time (average temporal timestep: 16.56 years, median timestep: 10 years). Three also include future vulnerability projections (Jurgilevich et al., 2021; Jongman et al., 2012; Giupponi et al., 2013).

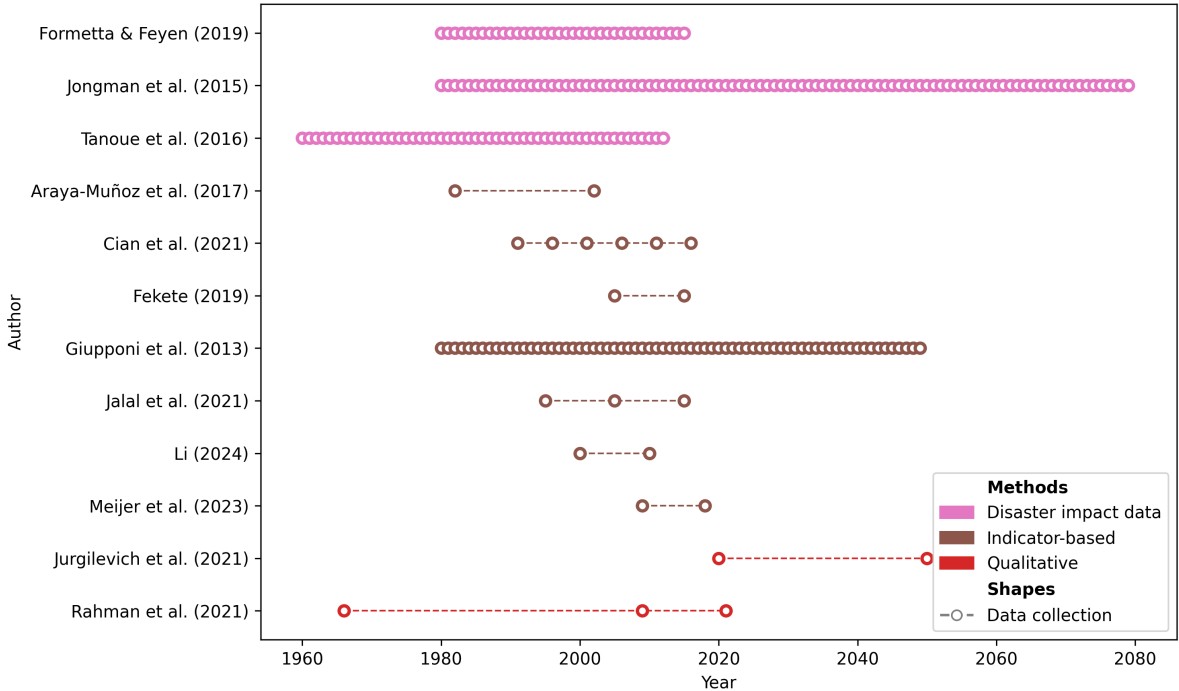

**Figure 6.** Analysis of underlying vulnerability dynamics at different points of data collection (circles) grouped by color based on the methods used. Note: Menoni et al. (2012) is not presented in this figure, as their analysis of underlying dynamics is oriented along the relative timing of different phases of the DRM cycle. Giupponi et al. (2013) uses a process model that assesses vulnerability on a yearly timestep. Formetta and Feyen (2019) and Jongman et al. (2012) use continuous impact data to determine the vulnerability dynamics on a yearly time step.

Indicator-based approaches are the most prominent methods (n=7) in this class of studies. They combine data from census 210 and cadastral records and earth observation data. A range of statistical methods is then used to reduce, standardize, weigh, and combine the underlying indicators into one index, including fuzzy logic modeling (Araya-Muñoz et al., 2017; Giupponi et al., 2013) simple and ordered weighting (Cian et al., 2021), Principal Component Analysis and Hierarchical Agglomerative Clustering (Meijer et al., 2022). Disaster impact data is used in two studies (e.g., Formetta and Feyen, 2019; Jongman et al., 2012).



They combined historic flood impact data with flood models to implicitly assess vulnerability by taking the relative number of
deaths compared to the total number of exposed people (mortality rate) and the relative amount of losses compared to the total
exposed GDP (loss rate). Lastly, studies applying qualitative analysis of underlying vulnerability dynamics most frequently re-
lied on literature and reports, complemented by cadastral and census data or interviews (Menoni et al., 2012; Jurgilevich et al.,
2017; Rahaman and Esraz-Ul-Zannat, 2021). Jongman et al. (2012), Formetta and Feyen (2019), and Tanoue et al. (2016)
determine vulnerability through loss/mortality rates, diverging from typical vulnerability assessment methods. All other stud-
ies consider multiple vulnerability characteristics. Qualitative studies (Jurgilevich et al., 2017; Rahman et al., 2021; Menoni
et al., 2012) cover a wider range of vulnerability subdimensions (around 5-9 subdimensions) than quantitative/indicator-based
studies (2-4 subdimensions), see Figure 4. All studies except Jongman et al. (2012) and Formetta and Feyen (2019) focus on
demographic and economic vulnerability drivers. Most also consider infrastructural/building vulnerability (Cian et al., 2021;
Jurgilevich et al., 2017; Rahaman and Esraz-Ul-Zannat, 2021; Fekete, 2019; Araya-Muñoz et al., 2017; Giupponi et al., 2013;
Menoni et al., 2012). Other subdimensions of vulnerability are rarely assessed, for instance, behavioral (Rahaman and Esraz-
Ul-Zannat, 2021; Araya-Muñoz et al., 2017; Menoni et al., 2012), environmental (Jalal et al., 2021; Rahman et al., 2021;
Giupponi et al., 2013), agricultural (Jalal et al., 2021; Rahman et al., 2021), governance/institutions (Jurgilevich et al., 2017;
Rahaman and Esraz-Ul-Zannat, 2021; Giupponi et al., 2013), awareness/information (Rahman et al., 2021; Menoni et al.,
2012), and health (Rahman et al., 2021).

## 3.5 Dataset for vulnerability dynamics

While investigating the data and methods used in different publications, we also collected a set of findings on the drivers and
consequences of dynamic vulnerability in these studies. We compiled a full list of these findings in the supplementary material
and briefly reflected on general patterns in the following.

Studies on **consecutive event dynamics** show that while flood exposure often leads to increased awareness and prepared-
ness, improvements are inconsistent, and psychological resilience does not necessarily follow, as vulnerability fluctuates due
to behavioral, cognitive, and structural factors. Phifer et al. (1988) found that flood vulnerability extends beyond immediate
damage, as health effects persist over time, particularly among those who experience both personal and community-wide de-
struction. Similarly, Köhler et al. (2023) identified a paradox where individuals with more flood experience tend to take more
precautionary measures but simultaneously feel less resilient. These findings underscore the role of psychological and social
dynamics in vulnerability. While Bubeck et al. (2012) demonstrated that flood events trigger accelerated mitigation efforts and
preparedness improvements, Kienzler et al. (2015) showed that these improvements are inconsistent across cases. The effec-
tiveness of early warnings and responses is highly dependent on flood characteristics and regional conditions. Schoppa et al.
(2024) expanded this understanding by modeling awareness and preparedness as dynamic processes influenced by emotions,
past experiences, and the deterioration of precautionary measures over time.

Analysis of paired flood events by Kreibich et al. (2017, 2023) reveals general trends across four vulnerability dimensions:
awareness, preparedness, emergency management, and coping capacity. Awareness often increases after flood events due to



experience or public campaigns, but cases such as Barcelona (1995–2018) show that improved access to information does not always translate into higher awareness. Preparedness typically improves through better forecasting and early warning systems, as seen in North Wales (1990–2013), but risk communication may remain ineffective, as in Barcelona. While emergency management structures improved in most cases (e.g., Beijing, 2012–2016; Danube Catchment, 2002–2013), coping capacity saw mixed results. While financial mechanisms such as insurance strengthened resilience in Kansas (1951–1993), other regions, such as Piura (1998–2017) and the Mekong (2000–2011), remained economically constrained and reliant on external aid. These patterns suggest that technical and institutional advancements alone are insufficient—long-term success depends on financial resources, governance structures, and community engagement. This stresses the importance of understanding and incorporating context in vulnerability assessments.

Studies on **co-occurring hazard dynamics** highlight that overlapping hazards can amplify vulnerability, shift risks between hazards, or even create new vulnerabilities through adaptation measures, demonstrating the need for integrated multi-hazard risk assessments. Bola Bosongo et al. (2014) demonstrated that crop production was significantly lower when floods and droughts co-occurred compared to droughts alone, reinforcing that overlapping hazards amplify negative outcomes. Similarly, Bernier and Padgett (2019) found that the combined impact of wind, wave, and water loads increased structural failure risk by 12%, emphasizing the importance of multi-hazard risk assessments.

Whytlaw et al. (2021) observed that evacuations during a pandemic introduced new vulnerabilities, such as financial insecurity and mental health challenges, revealing the need for adaptive strategies like expanding shelter locations and enhancing communication. Albulescu and Armaș (2024) applied an enhanced impact chain approach to multi-hazard conditions, showing that vulnerabilities can intensify, shift between hazards, or even emerge from adaptation measures. For example, evacuation strategies heightened infection risks, demonstrating how well-intended adaptation efforts can sometimes exacerbate vulnerability. These studies collectively highlight that failure to consider hazard interactions can lead to significant underestimations of risk.

Studies on **single-event dynamics** reveal that vulnerability manifests through socioeconomic disruptions, with floods exacerbating financial insecurity, altering migration patterns, and reshaping access to resources, leading to divergent recovery trajectories across different social groups. Salvucci and Santos (2020) used a Difference-in-Difference approach to show that floods caused a short-term consumption drop of 11–17% and an increase in poverty by six percentage points, illustrating how disasters deepen financial insecurity. Jamshed et al. (2020) examined post-flood changes in rural livelihoods, showing shifts in labor migration, credit access, and supply chains. Their findings highlight how rural households adjust to flood impacts by increasing their reliance on urban financial and informational networks, though poorer households with fewer social ties often struggle to access these resources. These results reinforce that vulnerability is deeply tied to socioeconomic positioning and infrastructure accessibility.





Studies on **underlying vulnerability** trends indicate that while economic development and policy interventions can reduce vulnerability over time, structural inequalities, demographic shifts, and unintended consequences of adaptation strategies continue to shape long-term risk dynamics. Araya-Muñoz et al. (2017) showed that poverty reduction led to decreased sensitivity, yet structural inequalities persisted, maintaining vulnerability disparities between regions. Fekete (2019) emphasized how demographic shifts, such as aging populations or infrastructure development, alter vulnerability landscapes, while Cian et al. (2021) demonstrated that economic and social transformations in the historic center of Padova pushed people to cheaper but less flood-prone areas, leaving this population less vulnerable.

Several global-scale studies revealed contrasting trends. Jongman et al. (2012) and Formetta and Feyen (2019) observed an overall decline in vulnerability due to economic development, particularly in lower-income countries. However, Tanoue et al. (2016) found that vulnerability follows an inverted U-shape, implying that economic growth does not always lead to linear reductions in risk. Rahman et al. (2021) demonstrated that infrastructure developments, such as embankments, initially reduce risk but later create new vulnerabilities, illustrating the unintended consequences of adaptation strategies. These findings confirm that vulnerability is highly dynamic and shaped by economic, social, and environmental interactions.

## 4   Discussion and Conclusion

In this study, we investigated methods and data for assessing vulnerability dynamics in the context of flood hazards, identifying key patterns, limitations, and gaps in the existing literature. The discussion is structured to address several core topics. First, we examine patterns in methods, data, and (sub)dimensions used across studies, highlighting dominant trends and challenges, specifically transferability and temporal resolution. Second, we explore methodological limitations, such as the reliance on static approaches, ambiguous terminology, and the lack of consensus on defining vulnerability. Third, we discuss the critical challenge of establishing causality in vulnerability dynamics research. Finally, we consider opportunities for future research, including the need for innovative approaches and the inclusion of overlooked vulnerability (sub)dimensions (Stolte et al., 2024). These discussions aim to provide a comprehensive understanding of the field's current state and outline pathways for advancing research on dynamic flood vulnerability.

### 4.1   Patterns across the vulnerability dynamics categories

Our review returned 28 publications with a strong representation of the underlying dynamics category. A somewhat unexpected finding was that we identified only two studies assessing vulnerability dynamics due to a single flood event. We expected a much higher number of studies addressing this dynamics category, as the analysis seems theoretically much easier and aligned with traditional flood impact assessment studies. A possible explanation is that the terminology used for this specific vulnerability dynamic category is different (e.g. using terms such as pre-event and post-event vulnerability, omitting the word "dynamic", see for example Moreira et al. (2021). Another explanation is that the data are not available in sufficient temporal resolution to investigate the single-event effects on vulnerability (e.g., comparing the building substance's state before and after the event).





The diverse set of methods, data, and scopes of the reviewed publications made comparing and identifying patterns complex
(see Figure 4 for a summary). One clear pattern we found is that most of the reviewed studies apply qualitative assessments
using forensics, narrative-based, or other descriptive approaches (e.g., Kreibich et al., 2017; Rahaman and Esraz-Ul-Zannat,
2021). This seems to be in divergence from traditional vulnerability assessments, which have much wider established applica-
tion of indicator- and curve-based vulnerability assessments (see, e.g., Nasiri et al., 2016; de Ruiter et al., 2017; Moreira et al.,
2021).


We find that indicator-based approaches are solely applied to assess underlying dynamics. These approaches often use
statistical methods to combine different vulnerability characteristics into indicators that can be merged into one index. However,
similar characteristics are used in multiple studies that apply statistical methods to analyze flood vulnerability dynamics (Köhler
et al., 2023; Kienzler et al., 2015; Bubeck et al., 2012; Phifer et al., 1988; Schoppa et al., 2024; Jamshed et al., 2020; Salvucci
and Tarp, 2021). However, none of these studies investigate how vulnerability indicators change because of (consecutive)
flood events. While the authors do not explain why they did not aggregate their statistical analysis of change into indicators,
we identify several possible reasons. Firstly, data like census and cadaster data usually come in less granular timesteps (i.e.,
annually or monthly), which may either not reflect changes due to flood impacts at all or obscure the effect of flood impacts
(see also the temporal resolutions in Figure 5 vs Figure 6). Secondly, many statistical methods studies investigated changes in
preparedness, awareness, and protection measures on a household scale. This information might be more valuable to directly
inform flood risk models (i.e., structural protection, behavioral changes) (Schlumberger et al., 2022). Thirdly, while indicators
are primarily used to describe a current state of vulnerability abstractly (Birkmann, 2007), trying to capture a change process
across multiple (potentially) dynamic vulnerability characteristics in one single value might be very challenging.

Only one study applies curve methods, despite this being a well-established method in traditional static flood vulnerability
assessment (e.g., Nasiri et al., 2016; van Ginkel et al., 2021; Arrighi et al., 2020; Fuchs et al., 2019). At a conference, Jochen
Schwarz and Holger Maiwald (2012) sketched the idea of the vulnerability curve of pre-damaged houses in the context of
consecutive flood and earthquake events. Still, we could not find any follow-up work implementing this idea. One possible
explanation for this limited application of curve methods for dynamic vulnerability might be that curve methods such as depth-
damage curves are not validated for static vulnerability assessments. Additionally, one could argue that the complexity of
embracing vulnerability dynamics makes it difficult to use such a quantitative, deterministic analysis of relevant processes for
objects that have more complicated (non-linear) failure modes than storage tanks (as in Bernier and Padgett, 2019). Dynamic
vulnerability curves are unavailable even for well-studied elements at risk, such as residential buildings. As a result, uncertain-
ties significantly increase when adding the complexity of dynamics and require a good understanding and validation of these
methods before being able to apply them in dynamic contexts.


Despite the recognized value of agent-based modeling approaches in assessing vulnerability (e.g., Taberna et al., 2020;
de Ruiter and van Loon, 2022) no peer-reviewed study seems to apply this method to the assessment of dynamics. During the
review process, we excluded a non-peer-reviewed study by Sobiech (2013) who developed and used empirical data from a case




study region in Northern Germany to initialize an Agent-based Model, where temporal vulnerability dynamics are introduced
using awareness, which is a function of time from the last flood event. Schoppa et al. (2024) and van Verseveld et al. (2015)
are the only two studies assessing dynamic vulnerability using process-based modeling approaches. An interesting approach
could be combining such approaches with methods such as those developed by Albulescu and Armaș (2024), who apply system
dynamics thinking to assess the effects of hazards and mitigation measures on the vulnerabilities within a system.

Regarding the (sub)dimensions of vulnerability taken into consideration, our analysis shows that most studies assess social
dimensions of vulnerability, particularly demographics, economic characteristics, awareness, and preparedness characteristics.
While being recognized as relevant, subdimensions of vulnerability, health, governance, and crime & conflict are the least
well represented (e.g., Matanó et al., 2022). While this finding is aligned with outcomes of other studies with regards to the
subdimensions of health and crime & conflict (e.g., Stolte et al., 2024), the lack of consideration of the governance element,
meaning, for instance, planning, empowerment, and (stakeholder) collaboration seems less intuitive. A possible explanation is
that most studies focus on vulnerability in the context of household-level risk mitigation or disaster response, where partially
practical institutional components such as emergency shelters are considered but not its organizational aspects.

Regarding the data types used for the assessment, interviews and surveys are the most frequently used data sources. Studies
assessing dynamic vulnerabilities of consecutive events relied only on interviews/survey data and reports, while all identified
data types were used in studies to determine underlying vulnerability dynamics. Some data types, such as census or cadaster
data, might appear less suitable for investigating vulnerability dynamics across different time scales due to their fixed, perennial
sampling intervals. However, other data types remain underexplored and could be highly appropriate for such assessments. For
example, earth-observation data, with its higher temporal resolution, could be combined with methods such as difference-in-
difference to investigate the vulnerability dynamics due to consecutive and single events(Bujis et al. *in preparation*).

**4.2   On the challenge of causality**

Despite the diversity of methods, data sources, and vulnerability dimensions used to assess dynamic vulnerability, a critical
challenge lies in addressing causality, as shown in Section 3.5. Most quantitative studies focus on correlation analysis, which
reveals associations between variables but falls short of identifying causal mechanisms. For instance, while studies such as
Phifer et al. (1988) and Fekete (2019) explore vulnerability dynamics, and others like Jongman et al. (2012) and Köhler et al.
(2023) discuss how changes in vulnerability alter impacts, there is a notable lack of research that pinpoints the causal pathways
of these dynamics. This gap limits the transferability and practical application of findings in disaster risk management. Con-
versely, qualitative approaches offer deeper insights into the processes driving vulnerability dynamics. Albulescu and Armaș
(2024) developed advanced impact chains to guide qualitative assessments. These chains map system-level processes that in-
fluence vulnerability, providing a structured way to conceptualize causality (*in review* Sparkes et al., 2024). Similarly, Whytlaw
et al. (2021) used expert knowledge to identify changes in vulnerability during co-occurring events, highlighting the complex
interactions that qualitative methods can uncover.



Assessing dynamic vulnerabilities in the context of flood hazards underscores these challenges. The goal extends beyond assessing risk as a function of hazard, exposure, and vulnerability to understanding how vulnerability dynamics evolve and affect risk reduction or emergency response measures (Mohammadi et al., 2024). For example, dimensions such as "underlying dynamics" and "co-occurring events" offer valuable insights into how vulnerabilities shift over time. Furthermore, analyses of consecutive events reveal how past flood experiences may influence future impacts, such as increased awareness or changes in preparedness at both individual and institutional levels (Köhler and Han, 2024). However, while these studies highlight essential relationships, they often remain descriptive and fail to establish causal mechanisms. Addressing the challenge of causality in dynamic vulnerability research is essential to advancing the field. Establishing causality would improve the reliability of findings and enhance their applicability to disaster risk reduction and management efforts.

### 4.3 The pitfalls of ambiguous terminology and other limitations of this study

The lack of consensus on vulnerability definitions (Kuhlicke et al., 2023; Rufat et al., 2019) posed a significant challenge for the analysis in this study. Divergent practices were found in attributing indicators to different social and physical dimensions of vulnerability or components of risk (hazard, exposure, vulnerability), using other concepts such as sensitivity and adaptive capacities. The term 'vulnerability' was inconsistently applied in distinction from the different components of risk. For example, we excluded a study by (Bryant et al., 2022), which examined how the Government of Alberta, Canada's optimized river operating rules affected flood risk, as the authors framed these changes as adjustments to hazard rather than vulnerability dynamics.

In addition, we found that the term "multi-hazard" was inconsistently applied. We identified numerous papers that use indicators that are relevant for multiple hazards (e.g., Quader et al., 2021; Faisal et al., 2021; Godfrey et al., 2015; Haque et al., 2020; Ghosh and Mistri, 2021; Mahadev and Rao, 2023; Mansour, 2019; Lazzati et al., 2023; Mullick et al., 2019), but do not account for the effects of multi-hazard interactions. As a result, they offer no insights into the impacts of multi-hazard dynamics or the spatio-temporal interactions between events. For example, Sarker and Adnan (2023) define tropical cyclones as multi-hazard events but use generic indicators (e.g., age group, poverty) to represent vulnerability and adaptive capacity without disentangling the relations between the different hazard-related impact drivers and these vulnerability characteristics.

Thus, the findings presented in this study should be interpreted in light of our methodological choices. As with any literature review, our results are biased by the search terminology used to identify publications. The search terminology was mainly built around 'dynamic vulnerability', a well-established term in the (multi-)risk community. At the same time, other research communities or previous research might use different terminology (see also the discussion of the limited representation of single-event dynamics). Terms such as "panel" and "longitudinal" are commonly associated with survey-based studies evaluating developments over longer time horizons in social science but may not have been captured by the search terms we used (Beauchemin and Schoumaker, 2016; Park, 2006). While a multi-step screening process and a double-review system ensured that we captured a broad and representative sample of studies relevant to dynamic flood vulnerability assessment, we did not adhere to formal review protocols such as PRISMA or SALSA. Instead, our method was designed to be flexible and exploratory,





reflecting the complexity and heterogeneity of the literature on dynamic vulnerability assessment for flood hazards. Decisions about relevance were made iteratively and collaboratively with the author team during the review process. While this approach introduced potential subjectivity in study selection, notably when excluding studies based on individual reviewer judgment, it also allowed us to review a large body of literature (i.e., 980 identified publications) on a highly heterogeneous topic. It

also enabled us to supplement or replace literature based on our knowledge of the field. For example, while the search terms returned Lan et al. (2021) as a potentially relevant study, we replaced it with Bernier and Padgett (2019), as this study includes the original vulnerability assessment of storage tanks that underpins the risk assessment undertaken by Lan et al. (2021). Nevertheless, our analysis should not be considered exhaustive, and methods and approaches from the non-peer-reviewed literature or published in languages other than English were intentionally excluded from our review.

**4.4 Advancing research on dynamic vulnerability: lessons and future directions**

We collected findings about the dynamics of vulnerability due to single-event dynamics, co-occurring or consecutive events, and underlying dynamics (see Section 3.5). Interestingly, many studies find a decrease in vulnerability in response to (a sequence of) events. A persistent challenge in vulnerability and risk assessment is the transferability of findings across different case studies (Kienzler et al., 2015; Moreira et al., 2021; Köhler and Han, 2024). For example, multiple studies on consecu-

tive event dynamics emphasize the role of prior flood experiences in reducing vulnerability. Conversely, Köhler et al. (2023) suggests that greater flood exposure does not necessarily enhance resilience. Instead, individuals may become better prepared while feeling less capable of coping. This indicates that adaptation is not solely a function of repeated flood exposure but is also shaped by cognitive and emotional responses, which may not be easily generalized across different population groups. Similarly, while specific broad trends - such as improved preparedness after flood events or the amplifying effect of multi-hazard

interactions - are commonly observed, the particular outcomes highly depend on local governance, socioeconomic conditions, and institutional capacities. This underscores that while dynamic vulnerability assessments can identify general trends, their predictive power remains limited unless they account for underlying structural conditions and potential unintended consequences of adaptation measures.

Co-occurring hazard studies further demonstrate that overlapping risks do not merely compound vulnerability but can also

shift unexpectedly. Albulescu and Armaș (2024) illustrate how floods and pandemics interacted to exacerbate vulnerability, with adaptation strategies, such as evacuation, unintentionally increasing the risk of infection. The ability to anticipate and manage such shifts depends on governance and institutional capacity, which varies significantly across contexts. Furthermore, methodological differences pose questions of comparability and transferability of the findings. For instance, studies differ in defining the temporal resolution of vulnerability dynamics. Köhler and Han (2024) examined how flood timing influences cop-

ing, while others suggest that risk perception is also shaped by the time elapsed since past events and active memory of past disasters (Lechowska, 2018). These differences call for further investigation into the sensitivies of temporal resolution of analysis and methodological approaches. Further, distinguishing between temporary adjustments and long-term transformations in vulnerability is a key challenge. While single-event studies such as Salvucci and Santos (2020) document immediate economic impacts, whether these short-term disruptions lead to sustained changes in coping capacity or vulnerability remains unclear.



Studies on underlying dynamics reveal contrasting long-term trends: Jongman et al. (2012) and Formetta and Feyen (2019) observed an overall decline in global vulnerability, while Tanoue et al. (2016) identified an inverted U-shaped trend, suggesting that economic growth may initially increase vulnerability before leading to reductions. These findings raise important questions about the sensitivity of vulnerability assessments to uncertainty and data availability

### 4.5 Broadening the scope of dynamic vulnerability research

In this study, we focus our review on flood-related vulnerability dynamics. Insights from research on vulnerability related to other natural hazards (e.g., earthquakes, droughts, or landslides) could be valuable inspiration for new methods applied to flood risk assessment (e.g., Cremen et al., 2022). In addition, we limited our review to studies that focus on vulnerability assessment (also about the search terms). Still, it may be useful to investigate how studies that assess flood impacts account for vulnerability dynamics. For example, Dottori et al. (2016) developed a flood impact model, INSYDE, that uses different vulnerability curves

based on the presence of different impact drivers within a single flood event (e.g., duration of flood event, consideration of water quality) and Schlumberger et al. (2022) made use of scenario-based impact assessments to account for potential changes/uncertainties regarding vulnerability. These studies provide examples of methods that could be used to investigate further how events increase/decrease the vulnerability of elements at risk, ultimately leading to vulnerability dynamics over time. We also see opportunities for expanding the utilization of agent-based modeling approaches (e.g., Thomson et al., 2023) or

exploratory modeling approaches (e.g., Moallemi et al., 2020; Schlumberger et al., 2024) to map processes and investigate the complex relationships between hazard, exposure, and vulnerability rather than predict their effects on vulnerability dynamics.

Finally, we observed some blind spots regarding the (sub)dimensions of vulnerability considered in the assessment methods, most notably crime and conflict, governance, and health. Thus, uncharted opportunities remain for new or tailored methods for more holistic assessments of dynamic vulnerability.

*Data availability.* The list of reviewed publications and the dataset of the identified findings regarding the drivers and effects of dynamic vulnerability will be made available on Zenodo upon publication.

*Author contributions.* We use CRediT to distinguish authors' contribution. *Conceptualization*: PW (equal), MdR (equal), RST (equal). *Data Curation*: JS (lead). *Investigation*: AS (equal), HG (equal), JS (equal), MdB (equal), TS (equal), WJ (equal), AT (supporting), PW (supporting). *Methodology*: JS (lead), MdB (equal), AS (supporting), HG (supporting), TS (supporting), WJ (supporting). *Formal Analysis*: JS
(lead), HG (equal), AS (equal), TS (equal), WJ (equal). *Visualization*: JS (lead), TS (supporting). *Writing – Original Draft*: HG (equal), JS (equal), MdB (equal), TS (equal), WJ (equal). *Writing – Review & Editing*: JS (lead), AS (equal), MdB (equal), PW (equal), TS (equal), HG (supporting), MdR (supporting), RST (supporting).



*Competing interests.* The authors have the following competing interests: Antonia Sebastian, Marleen de Ruiter and Robert Šakić Trogrlić are editors of the Special Issue we are submitting this manuscript to. Also, Robert Šakić Trogrlić and Philip Ward are editors for NHESS.

*Acknowledgements.* JS, TS, WJ, PW, MCdR, and RST received support from the MYRIAD-EU project, which received funding from the European Union's Horizon 2020 research and innovation program under grant agreement No. 101003276. This work contributes to the goals of the IAHS scientific decade of HELPING (2023-2032).



# Appendix A: Categories used to classify the analyzed publications

**Table A1.** Categories used to classify the analysed publications

| Categories | Options | Typology source |
|---|---|---|
| Approach | (a) Indicator-based, (b) Curve, (c) Process-based modeling, (d) Statistical analysis, (e) Disaster impact data (f)Qualitative analysis | Extended based on Nasiri et al. (2016) |
| Dynamic vulnerability category | (a) single-event dynamics, (b) consecutive event dynamics, (c) co-occurring event dynamics, (d) underlying dynamics | Extended based on de Ruiter and van Loon (2022) |
| Elaboration on how temporal and multi-hazard dynamics are considered | Free-text elaboration | |
| Description of the method(s) used | Free-text elaboration | |
| Data | (a) Building and infrastructure data; (b) Damage/impact data; (c) Earth observation; (d) Focus groups/workshops; (e) Fragility model; (f) Hazard data; (g) Interviews/Surveys/Questionnaires; (i) Regional plans; (j) Reports/documents; (k) Socioeconomic data | Developed based on the authors' expertise |
| Data source | (a) Cadastral Data; (b) Census; (c) Field Monitoring Data; (d) Interviews etc.; (e) Literature and Reports; (f) Maps and Topography; (g) Modelled Data; (h) Remote Sensing Data; (i) Workshops etc. | Developed based on the authors' expertise |
| Social vulnerability categories | (a) Awareness & Information; (b) Crime & Conflict; (c) Culture & Behavior; (d) Demographic; (e) Economic; (f) Governance; (g) Health; (h) Institutional | Stolte et al. (2024) |
| Physical vulnerability categories | (a) Critical Infrastructure; (b) Environment; (c) General (urban) assets | Stolte et al. (2024) |
| Scale | (a) global, (b) regional, (c) sub-national, (d) national, (e) local, (f) multiple scales | |
| Case study or review paper | (a) yes (b) no | |



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
