# Peer review of "Review article: Stocktaking of methods for assessing dynamic vulnerability in the context of flood hazard research"

_EGUsphere, 2025_

## Referee Comment (RC1)

**Specific comments and technical corrections of "Stocktaking of methods for assessing dynamic vulnerability in the context of flood hazard research"**

Overview

This article aims to systematically review methods used to assess dynamic vulnerability in the context of floods. The authors also state that they compiled their findings about the drivers and effects of vulnerability dynamics in a dataset (but do not make this available to the reviewer as far as I can tell). The concept of dynamic vulnerability is important for many use cases in both research and practice. Unfortunately, the manuscript is unorganized, often vague, and many of the claims are imprecise. There are signs in the writing that the authors did not finish revising and editing the paper (e.g., an incomplete sentence L180-181 among other sloppy instances). These are limitations that a careful revision could overcome if the editor was forgiving. However, the review is not systematic and that is the main potential value of the manuscript. To overcome this challenge, the review needs a more rigorous sampling strategy, more distinctive and clear conceptual classifications, and an insights-driven synthesis approach.

Section Comments

Abstract

- Difficult to parse because of vague language choices and a lack of examples. For example, the authors don't define "dynamic vulnerability." They explain that dynamic vulnerability is driven by a wide range of characteristics and that it is critical to understand flood risk.
- The abstract should define key concepts, such as dynamic vulnerability. Perhaps the authors feel that the term is self-evident or widely understood, but it is ambiguous to me and their sample of 28 studies suggests it is not a widespread concept. The second paragraph of the Intro suggests that the authors recognize vulnerability is a pluralistic concept and needs a definition.
- The authors should consider whether their four categories of vulnerability dynamics are concepts that can be grouped together (the events categories probably can be, but "underlying dynamics" seems like a different conceptual entity) and whether it is appropriate to call them "vulnerability dynamics."
- It is difficult to know what this study is about by reading the abstract.
- It seems unfair to highlight in the abstract that a key methodological gap is the limited integration for multi-hazard. The title and beginning of the abstract explicitly frames the study "in the context of flood hazard research."

Introduction

- Imprecise writing. A few early examples:
    - What do the authors mean by "robust protective infrastructure?"
    - The first paragraph of the introduction talks about the *number of flood events reported* in a database. The following sentences are about "this increase." Do the authors really want to focus on reported events in a database? It seems like with the reference to vulnerability (also the central framing of the paper) they mean to talk about an increase in flood impacts over time?
- L41 – "These challenges are evident in assessments of…" -> how? It would be helpful to have an illustration of how the challenges are evident instead of just saying they are.
- L46 – I don't know what the authors mean by the "what" of assessments or the "why" and "how." Do they mean vulnerability in "what," "how" and "why?" Please be more specific. The following examples do not clarify the what/why/how distinctions. I'm not sure how to place the subsequent examples in the context of the preceding sentences. This is the scoping paragraph of the paper. If the reader does not understand these points, they will struggle to understand the purpose or contributions of this paper. The title and abstract make the review seem focused on methods of accounting for or measuring dynamic vulnerability, but this scoping paragraph seems disconnected from that framing.
- The authors cite Simpson et al., 2021 but make no mention of their proposed "response" dimension that has an implicit dynamic component to it. How can the authors reconcile the "response" dimension to complex risk assessment in that study with their "dynamic vulnerability" framing here? I'm sure that is possible to do, but it is not done here. I think it is necessary to do because these definitions are fickle and the article presents itself as a review, so should aim to be comprehensive and to clarify muddled, overlapping, and imprecise concepts in the field.
- L52 – "a comprehensive overview of approaches for assessing dynamic flood vulnerability" in reviewing only 28 studies? This is not a good sign. I'm worried the authors are too attached to this "dynamic vulnerability" language choice instead of studying the underlying phenomena and processes required to develop a rich sample and understanding. Many fields like economics and sociology study things that can fit into "vulnerability" but may not use that language choice. One reason is that, as the authors indicate, vulnerability is multifaceted and pluralistic so some fields like to be more specific about what they are studying than using a vague catch-all term. Another reason is that there

is increasing attention in some fields to avoiding stigmatizing language such as "vulnerability" - https://www.nih.gov/about-nih/what-we-do/science-health-public-trust/perspectives/writing-respectfully-person-first-identity-first-language.

Methods

- Section 2.1: I'm not convinced these are "types" of "categories" of dynamic vulnerability. A "type" or "category" of vulnerability might be "financial vulnerability." The choice to call these concepts "categories of dynamic vulnerability" is a major concern for me. I understand that the authors base this on a previous paper, but that does not mean it is a useful conceptual device for classifying approaches. Figure 1 makes panels 1-4 seem like the response of vulnerability to different exogenous events (sometimes flood-related, sometimes not). Not sure that characterizing the sequence of events which the icons/colors represent) is a "category" of dynamic vulnerability because there can be so much heterogeneity in the vulnerability to response to any event or sequence of events. It's such a limited representation of the dynamics surrounding agent decision-making that it's distracting. For example, if a person loses their job, they are most likely more susceptible to some forms of harm from flooding than they were before. If a person gets a raise, they are potentially less susceptible to some forms of harm from flooding than they were before. For instance, they could directly cope with economic flood damage or invest in protective infrastructure. But they could also increase their spending and take on more credit, possibly having a lower budget or financial capacity to deal with harms associated with flooding. In summary, I think "vulnerability dynamics" can refer to the change in vulnerability surrounding any of the individual exogenous events or an entire span of events. I don't see how the number and type of events are different categories of vulnerability dynamics. They seem more like different realizations of exogenous events that different people/organizations/institutions may respond differently to.
- More on Figure 1 - I think it is attractive and promising, but the icons on the top are not self-evident, there is no link to the definitions of the categories and the 4 panels (is it supposed to be 1:1?), I'm not sure what is the purpose of the magnifying glasses, and the lack of a legend makes aspects uninterpretable. I think part of the problem is how abstract the concepts of vulnerability and dynamic vulnerability still are at this point of the manuscript. The Figure 1 might work better if it sticks to a specific example, like a structure's susceptibility to damage for a fixed level of flooding, over different sequences of shocks. It might work to have a few rows with different examples, demonstrating how vulnerability

dynamics look different for different types of actors and different sequences of events/shocks.

- The concept of vulnerability is so abstract here and it would be very helpful to have more concrete examples. Otherwise, it is ambiguous and distracting to keep seeing references like "financial vulnerability." What does this mean? I think the reader needs guidance on a specific example of a vulnerability and how it can change over time. On the financial vulnerability reference, I have a lot of confusion. Referring to Thomson et al. 2023, they write "[Our] approach utilizes a series of … to improve understanding of how systemic financial risk could arise from flood impacts to residential properties. As such, this work illustrates a more nuanced approach to evaluating flood-induced financial vulnerabilities." I know this is not the paper under review, but the present study characterizes Thomson as assessing "financial vulnerability" so I wanted to raise the relevant context. Is systemic financial risk a type of flood-induced financial vulnerability? Perhaps, but it's not self-evident and needs much more development and spoon-feeding as a concept. What is the element at risk? What are the specific social, economic, and physical characteristics of the element at risk that characterize "financial vulnerability?" Why isn't "financial vulnerability" a vulnerability category? Are we just talking about income?

- I don't see the four categories of dynamic vulnerability as a useful division of concepts. The examples for co-occurring event dynamics and underlying dynamics categories seem very similar. For example, a pandemic lockdown can induce an economic crisis. I think the authors need a better classification scheme or much more justification for the one proposed here. On L59 they say "we identify four categories of dynamic vulnerability essential for flood risk assessment (Figure 1)" but don't say *how* they identified those categories.

- The authors claim the review is systematic (L81) but it's not clear how to determine what a successful systematic review "of dynamic flood vulnerability" looks like. Is it models of dynamic flood vulnerability? Is it drivers? Is it data sources for evaluating? The authors may feel that my subsequent comments miss the mark because I focus on things outside their scope, but the issue is that they don't define their scope clearly in the Introduction. "Approaches for assessing dynamic flood vulnerability" is general and vague, yet only 28 studies in the sample? And I see that none of the studies that immediately come to my mind (anchored in the Thomson example thinking about "financial vulnerability" which I think is really just income) are cited in this paper, like:
  - Gallagher, Justin, and Daniel Hartley. 2017. "Household Finance after a Natural Disaster: The Case of Hurricane Katrina." American Economic Journal: Economic Policy 9 (3): 199–228.

- o Deryugina, Tatyana, Laura Kawano, and Steven Levitt. 2018. "The Economic Impact of Hurricane Katrina on Its Victims: Evidence from Individual Tax Returns." American Economic Journal: Applied Economics 10 (2): 202–33.
  - o Deryugina, Tatyana. 2017. "The Fiscal Cost of Hurricanes: Disaster Aid versus Social Insurance." American Economic Journal: Economic Policy 9 (3): 168–98.
  - o Or even Kousky, C., Palim, M., & Pan, Y. (2020). Flood damage and mortgage credit risk: A case study of Hurricane Harvey. Journal of Housing Research, 29(sup1), S86-S120 which is a motivating reference for Thomson et al. (2023).
- Also on the systematic claim, I don't understand how investments in flood-risk reduction are out of scope of a review focused on dynamic vulnerability. These investments often occur in the context of post-disaster intergovernmental transfers and disaster aid. For example, see Deryugina, Tatyana. 2017. "The Fiscal Cost of Hurricanes: Disaster Aid versus Social Insurance." American Economic Journal: Economic Policy 9 (3): 168–98 and Davlasheridze, Meri, Karen Fisher-Vanden, and H. Allen Klaiber. "The effects of adaptation measures on hurricane induced property losses: Which FEMA investments have the highest returns?." Journal of Environmental Economics and Management 81 (2017): 93-114. This is also a particularly interesting dynamic because of connections to gentrification. Areas that become safer may become more desirable to live, attracting investment and increasing overall exposure (and potentially risk!) while also potentially displacing people with lower incomes.
- I am surprised to see that Google Scholar was used for obtaining an initial sample. A recent article in PlosOne recommends in its abstract that "whilst Google Scholar can find much grey literature and specific, known studies, it should not be used alone for systematic review searches. Rather, it forms a powerful addition to other traditional search methods." See Haddaway, Neal Robert, et al. "The role of Google Scholar in evidence reviews and its applicability to grey literature searching." PloS one 10.9 (2015): e0138237. One of the most important reasons not to use Google Scholar is that it is much more difficult to reproduce and replicate a search than using a more typical database for a systematic review such as Scopus or Web of Science. The large number of not peer-reviewed studies in their initial sample and studies excluded based on title and abstract make the use of this search engine questionable as well.
- In addition to the use of Google Scholar, I find several aspects of the article sample process flawed:
  - o Criterion iii requires that the article be freely accessible to the reviewers. Was interlibrary loan not available to the authors? This can unnecessarily

limit the sample size and with authors from so many institutions it is surprising that this is a limiting factor for including some studies. Dropping 35 studies for this reason seems

- o Criterion v requires that the study adopt a definition of vulnerability consistent with the IPCC or UNDRR. This seems to substantially limit the authors' ability to systematically review dynamic vulnerability and how it is studied in research. It seems it would be more effective to search for research that focuses on *specific* drivers of dynamic vulnerability (like recovery, changes in income, adaptation funding, infrastructure projects, etc.,). This criterion seems to induce a selection bias for disciplines that utilize vocabulary preferred by the authors. I think the review would have more useful insights if it expanded its search to include studies that investigate

- o Criterion vi requires that the study address one of the vulnerability dynamics identified in Figure 1. Seeing that these dynamics were *a priori* determined by the authors, instead of informed by a review of potentially relevant literature, is concerning and demands that the authors more transparently define *how* they identified these categories. They do not do this in the beginning of section 2.1.

- o Criterion vii requires that the study provide details allowing replication. This is a major limiting factor. What degree of published studies adequately provide details allowing replication? Few studies in the area of flood risk assessment even make their data available, a requirement for replication.

- o Criterion viii requires that the study is a case study. Why? What is an example of a study that is not a case study and was excluded for that reason?

- All of the criteria for screening need a justification, and many of them lack one.
- How many of the studies in the final sample of 28 come from the additional papers added by collaborators?
- On Line 98 they say they categorize methods into five groups but then proceed to only define four groups.

Results

- I am skeptical about their Google Scholar search. The first study mentioned in the results section is Phifer et al., 1988, which must be added by one of the authors because it has no mention of "vulnerability assessment" or "vulnerability analysis" and these are requirements for search terms. If the first relevant study

does not have these search terms, why do the authors think these are reliable search terms to guide a "systematic" search? The authors would likely better serve their review goal if they started from their known examples of studies and worked backwards to identify search terms.

- The Phifer et al., 1988 article raises another issue – is this article about vulnerability or impacts? The title is "The *Impact* [emphasis mine] of Natural Disaster on the Health of Older Adults: A Multiwave Prospective Study" and in the abstract states "The present study examined the *impact* [emphasis mine] of flood exposure on the physical health of this sample..." Revising the IPCC definition, "vulnerability refers to the social, economic, and physical characteristics of an element at risk that make it susceptible to harm in the event of exposure to a hazard." I suppose the element at risk here is the health of older adults. In that case, how is the study's focus not about the element at risk of health of older adults? One could argue that declining health can make one more susceptible to future declines in health. This raises the concern that the authors of the review are not clear about vulnerability "in what" and the review can complicate the concept of vulnerability in the reader's mind instead of clarifying and enriching it.

- The relatively high proportion of studies in Europe raises concerns about the sample of studies as well, given the high representation of European authors. Given the content and focus of the Phifer et al., 1988 article, I can think of a number of health and economic studies that are appropriate for the authors to include (such as those I cited earlier). I think the authors would better serve their "systematic" review goals if they used a snowball sampling approach from the studies they knew, looked at citations/references for relevant studies to include, and then worked backwards from this sample to identify appropriate screening terms for a broader review of relevant articles.

- The idea that only two studies assess vulnerability dynamics due to single flood events is wrong. I know this because one of the studies they describe in the single event section investigates the impact of the 2015 Mozambique flood on household consumption and poverty levels, using a difference-in-difference approach "to quantify changes in vulnerability attributable to the flood event." If income, poverty, welfare, etc., are all forms of vulnerability (this was not clear to me as there was no description before on how one can measure vulnerability), then this review is missing a sizable number of studies in economics that use a wide variety of methods in causal inference to evaluate the impact of floods on households. In addition to the studies I cited before, there are a number of studies that evaluate how disasters and other exogenous events (like risk communication efforts) affect flood insurance take-up and flood-risk reduction investments. The review misses key studies from authors such as Kunreuther, Kousky, Czajkowski, Botzen, Bubeck, and Gallagher (and these are authors I

know without dynamic vulnerability being a specific area of focus for me – surely they miss many more key studies and authors). For example:

- o Bubeck, P., Berghäuser, L., Hudson, P., & Thieken, A. H. (2020). Using panel data to understand the dynamics of human behavior in response to flooding. Risk Analysis, 40(11), 2340-2359.
- o Bubeck, P., Botzen, W. J., Kreibich, H., & Aerts, J. C. (2013). Detailed insights into the influence of flood-coping appraisals on mitigation behaviour. Global environmental change, 23(5), 1327-1338.
- o Botzen, W. W., Kunreuther, H., Czajkowski, J., & de Moel, H. (2019). Adoption of individual flood damage mitigation measures in New York City: An extension of protection motivation theory. Risk analysis, 39(10), 2143-2159.
- o Gallagher, J. (2014). Learning about an infrequent event: Evidence from flood insurance take-up in the United States. American Economic Journal: Applied Economics, 206-233
- o Kousky, C. (2017). Disasters as learning experiences or disasters as policy opportunities? Examining flood insurance purchases after hurricanes. Risk analysis, 37(3), 517-530

- The results section describes studies and characterizes their features (e.g., in Figure 4) in ways that do not convey insights. They hardly break down Figure 4 in the results text. Part of the problem is that the review does not state clear research questions so some of the results seem non sequitur. For example, starting on L154, the authors discuss the timing of data collection. They discuss this for 4 sentences. This is in a paragraph about "the time intervals between the consecutive events, the number of flood events considered, and the duration of the analyses." With this paragraph's stated focus, why is it "important to note that Kreibech et al. (2017, 2023) use a literature review considering various reports and publications from different years to assess vulnerability dynamics, while we took the date of the scientific publication to determine the time lag between events and data collection?" This long discussion seems irrelevant. What does any of this have to do with vulnerability dynamics? What does it have to do with methods for assessing vulnerability dynamics?

- L179: "The studies mentioned above capture the vulnerability dynamics regarding different elements at risk and vulnerability dimensions. While Phifer et al. (1988) focus on human health and well-being, taking into account demographic, economic, and health dimensions of vulnerability," -> that's the end of the sentence! This is unacceptable for a submission to such an esteemed journal.

- L196 – this paragraph's focus on elements at risk and contributing vulnerability dimensions is shallow and vague. It would be much more helpful for the authors

to explicitly state what the vulnerability dimensions are and what the impact of floods are on these dimensions. For example, what does the following mean? "Albulescu and Armas (2024) use augmented impact chains to express the effects of hazard impacts and risk mitigation measures on vulnerability without focusing on a certain element at risk"

- Section 3.5 is titled "dataset for vulnerability dynamics" but then the text on 235 starts talking about findings from studies. Very unorganized and confusing. It's nice to see some discussion of the findings from these studies, but it is presented in an unorganized way and is hard to follow. The separation into "categories" of consecutive event dynamics, co-occurring hazard dynamics, single-event dynamics, and underlying vulnerability detracts from the flow of the discussion. This separation forces distinctions that move relevant findings apart from each other. Review articles should synthesize in ways that produce insights, not just produce a list of thematically linked findings and leave it to the reader to connect the dots.

Discussion

- I don't understand how the indicator-based approaches, which the authors state entirely avoid investigating how vulnerability indicators change because of flood events, can be included in this review on vulnerability dynamics. This again supports the idea that the authors' categories of vulnerability dynamics are not helpful for synthesizing vulnerability dynamics. In fact, their categories are more like hazard dynamics (these are mostly hazard realizations over time). Vulnerability dynamics suggest attention to changes in vulnerability dimensions over time.

- The discussion on "curve methods" L334 is underwhelming. The discussion does not acknowledge a wide literature on flood damage functions and substantial uncertainty in relating flood and structure characteristics to economic damages. There is also a random seeming reference to "storage tanks" in L341. What exactly do the authors mean that a damage curve is static? I think they mean that most damage curves take generally static characteristics of a structure as input, such as number of stories, foundation type, and first-floor elevation, and produce percent damage estimates for possible inundation depths relative to first floor. But that's not all damage functions, and even these characteristics can change over time. For example, consider the following study that uses building quality as an input variable, a structure characteristic that may change between flooding at time t1 and time t2 if the damage is large enough: Schröter, K., Kreibich, H., Vogel, K., Riggelsen, C., Scherbaum, F., & Merz, B. (2014). How useful are complex flood damage models? Water Resources Research, 50(4), 3378-3395.

This topic is complex and would benefit from more thoughtful consideration of relevant literature.

- Why "agent-based modeling" on L346 but "Agent-based Model" on L349?
- How can the authors claim that "no peer-reviewed study seems to apply [agent-based modeling] to the assessment of dynamics?" L346. For example, see Taberna, A., Filatova, T., Hadjimichael, A., & Noll, B. (2023). Uncertainty in boundedly rational household adaptation to environmental shocks. Proceedings of the National Academy of Sciences, 121(29), e2215675120. I am sure there are many more relevant agent-based modeling studies.
- The discussion on causality is welcome, but underwhelming. The authors miss many economic studies that use methods in causal inference to identify the effects of floods and other exogenous events on features the authors defined elsewhere as vulnerability (see my many references to economics papers above). The main challenge is the transferability of these findings to the many analytical contexts in which projections of vulnerability dynamics can help inform risk assessments and decisions. Such a discussion is extremely important and would be a very nice contribution to the literature.
- I am confused that the authors highlight that they excluded a study by Bryant et al., 2022 because Bryant et al. frame changes as adjustments to hazard rather than vulnerability dynamics. Making a choice based on language rather than whether the object of study is the same is a poor way to do the kind of synthesis this "systematic review" aims for. Committing to strict language choices for a new idea of "vulnerability dynamics" – which is not widespread or standardized (like the way the IPCC is a helpful standard definition for "vulnerability") – is clearly a limiting factor of this review. There are clearly many studies that evaluate changes in vulnerability over time. Leaving these out because of language choices – as opposed to connecting relevant fields and literatures to create a richer concept of dynamic vulnerability (if that's even the appropriate catch-all for what the authors want to synthesize) – is a failure to be systematic.
- I appreciate their discussion on limitations, but I disagree with the dismissal of these limitations as a normal part of any review and disagree that their process "ensured that we captured a broad and representative sample of studies relevant to dynamic flood vulnerability assessment." (L414) I think the authors miss too many relevant references to consider the review systematic or helpful. The authors are clearly aware that terms such as "panel" and "longitudinal" would be helpful and don't satisfyingly justify why they stick to jargon in their field of (multi-) risk. The topic of study is important and for publication in an interdisciplinary journal such as this one, it is necessary that the authors take efforts to increase representation of studies outside of their own small community.

---

## Referee Comment (RC2)

Overall comment:

The study aims at reviewing the methods, contents and datasets of dynamic vulnerability assessments to floods, while basing on the previous conceptualizations of vulnerability dynamics. While the study offers some interesting insights in terms of methodological development, it has two major drawbacks: 1) unjustified methodological choices, that led to a small and possibly very limited sample, and 2) novelty – while the study points out gaps, and claims to provide "roadmap for advancing more robust and dynamic flood vulnerability assessments", it stops short of that and focuses mainly on reiterating what has or has not been done. In sum, the study could be worthy of publication if it a) fulfilled a proper systematic search and review strategy, which is in this case doable and warranted; b) reviewed an exhaustive sample of papers, c) produced a bit more interesting contribution beyond gaps.

Comments in more detail:

1) The gap that the study addresses could be articulated more clearly – p. 2 lines 45-50 – the references to vulnerability of what and "how" and "why" could be opened up. If the authors refer to the methods ("how?") then Jurgilevich et al 2017 review covers that, in addition to what. I'm not sure what authors understand as to "why".

2) It would be beneficial if the authors could explain as to why we need to understand/assess V dynamics from the perspective of multi, cascading and aggravating hazards. It is somewhat articulated that vulnerabilities can be "interacting", btu more tangible substantiating examples would be beneficial. Also, a lot of vulnerability indicators, drivers or dimensions are the same for several hazards (e.g., typical indicators such as age, income, housing type, education level are relevant to consider for floods as well as for heat-related events, storms and others), so why do we need to account for them separately in e.g. cases of consecutive events?

3) Is the overall rationale that vulnerability is also driven by impact, so vulnerability is dynamic as a result of a hazard in addition to its own inherent dynamics? Isn't this what is called dynamic risk?

4) Line 55 – if the study points our gaps and provides synthesis – it is not a roadmap. The actual contribution of the paper stops at synthesizing gaps and advances

5) Main criticism concerns methods- I do not consider methodological choices of the authors justified enough not to adhere to the protocol of systematic review. The research question of the study warrants a systematic review, and the field is homogenous enough to pursue it (as previous reviews have done successfully). The search keywords (flood, vulnerability) position the sample firmly into risk and adaptation literature, thus there is little challenge of dealing with the definition of vulnerability from epidemiology for example. In this vein, the justification of following a semi-systematic review is weak.

6) Google Scholar is not an appropriate search engine as it is guided by algorithms and previous user history.

7) Furthermore, the authors have a very limited search sequence. For example, search for risk assessments could yield more suitable papers, as these which often contain the assessment of vulnerability. Additionally, vulnerability dynamics may be an established term in a very niche theme of multi-hazard research, but it is not a well-established term overall, and there are plenty of studies that do relevant things but do not call it vulnerability dynamics. Overall, the search sequence limits the sample in many ways.

8) The authors have a limiting set of inclusion criteria. For example, criteria 5 – the study has to adopt a definition of vulnerability from AR6 is too restricting, because IPCC SREX and AR5 have pretty much the same definition and conceptualization. Even if the conceptualizations in the studies are somewhat different stemming from the evolution of vulnerability concept (in IPCC AR4, 5 and 6) – it is still a doable task to appraise the literature and categorize according to the definitions adopted in this study (AR6).

9) Lines 90-95 discuss why previous IPCC AR definitions have been excluded. I will challenge this rationale – previous studies can indeed be comparable, where vulnerability as per AR5/6 corresponds to adaptive capacity + sensitivity. Previous reviews have done it that way, and there is plenty of studies that point out the evolution of risk and vulnerability frameworks where AR4 vulnerability is comparable to AR5/6 risk, and AR576 vulnerability is adaptive capacity + sensitivity in later frameworks.

10) The study has a very extensive section on limitations, mainly justifying their methodological choices and treating many of these as inherent review choices. The comments above highlight that many if these limitations could have been overcome and some choices are not justified enough.

---

## Author Comment (AC2)

The study aims at reviewing the methods, contents and datasets of dynamic vulnerability assessments to floods, while basing on the previous conceptualizations of vulnerability dynamics. While the study offers some interesting insights in terms of methodological development, it has two major drawbacks: 1) unjustified methodological choices, that led to a small and possibly very limited sample, and 2) novelty – while the study points out gaps, and claims to provide "roadmap for advancing more robust and dynamic flood vulnerability assessments", it stops short of that and focuses mainly on reiterating what has or has not been done. In sum, the study could be worthy of publication if it a) fulfilled a proper systematic search and review strategy, which is in this case doable and warranted; b) reviewed an exhaustive sample of papers, c) produced a bit more interesting contribution beyond gaps.

Response:  We want to thank the reviewer for the extensive review provided. We appreciate the time this reviewer took to offer their reflection on the paper, and we intend to use this review to sharpen the paper's scope and address concerns about its scientific relevance. As we understand, the main critiques relate to a mismatch between the planned scope of the article and the methodological choices that were made, along with the reviewer's confusion about the objective of the article  stemming from an insufficient elaboration on said scope and the key terminology around vulnerability and dynamics. We understand the reviewer's concerns and have reflected on several options that we hope help to address them.

We acknowledge the ambiguity in the introduced concepts and categories used for the analysis. We intend to better contextualize this research by defining the term of vulnerability and articulating how we are using it in our review, along with the term 'vulnerability dynamics'. For the latter, we don't want to use earlier categorizations (e.g., based on de Ruiter & van Loon, 2023) as a starting point but as a subject for investigation and thus a potential outcome.

Second, we acknowledge that despite our initial claim, the research conducted cannot be called systematic. We discussed options to redo our search by including Scopus and/or Web of Science. However, the original search terms do not return sufficient substance on either search platform, and would necessitate an entirely different strategy for querying these databases and thus require starting from scratch. While we acknowledge that the paper requires significant revisions, we do not believe that re-structuring our entire literature search is necessary to address the reviewer's comments. Instead, we propose that we would more clearly articulate that the purpose of this manuscript is to showcase methods that have been or could be used to assess dynamic vulnerability in the context of flood-related hazards and their impacts. More specifically, we investigate how conventional vulnerability assessment methods (e.g., social vulnerability indices, physics- and process-based models, and statistical and narrative-based methods) can be used or tailored to assess vulnerability dynamics. Instead of using a set of vulnerability categories (as done in the original version), we intend to reorganize the existing material to focus on the methods themselves. We specifically investigate what data are used, which (sub-)dimensions of vulnerability are considered, how these methods capture temporal dimensions of vulnerability dynamics (time span and resolution), or not, and what changes were investigated using these methods, if any. In restructuring the manuscript to achieve this goal, we can leverage all of the papers we have already collected and reviewed as part of our initial submission.

Third, we intend to discuss whether there are any patterns visible regarding specific causes or situations that result in vulnerability dynamics. This should offer some evidence to reflect on possible ways to categorize vulnerability dynamics, also in the light of already proposed conventions (e.g. de Ruiter & Van Loon, 2023). We do not want to create one based on the evidence we collect, but hope it could be a starting point for future attempts to come up with a relevant categorization of different vulnerability dynamics. Lastly, we also want to articulate limitations of the methods introduced in the papers that we included in our review.

We believe such an analysis of the existing literature still has significance, even if it is not structured as a systematic review. As the reviewer noted, vulnerability research is vast and (partly) ambiguous. Vulnerability dynamics are a subject of interest at the moment, but there are limited insights into how to best assess vulnerability dynamics and their contribution to risk profiles or outcomes. With our review, we want to showcase the existing research in this area, focusing on what has been used/produced in relation to the flood (disaster) risk management community.

We are convinced that the data we have collected offers a promising starting point for providing relevant insights to the community. However, acknowledging the methodological limitations of our current approach, we apply a targeted literature review to identify relevant studies across different methodological frameworks. We intend to use a similar approach to what has been done by Ward et al. (2020; https://doi.org/10.1016/j.wasec.2020.100070) and di Angeli et al. (2022; https://doi.org/10.1016/j.ijdrr.2022.102829), which are just two of many examples of influential research using targeted (non-systematic) literature reviews. With these changes to our analysis and intended scope , we offer a starting point for further investigations from within the flood risk management community, which can be linked to investigations from other hazard communities (e.g., fire, drought) in the future.

Below, we include line-by-line responses to the reviewer's comments.

Comments in more detail:

1) The gap that the study addresses could be articulated more clearly – p. 2 lines 45-50 – the references to vulnerability of what and "how" and "why" could be opened up. If the authors refer to the methods ("how?") then Jurgilevich et al 2017 review covers that, in addition to what. I'm not sure what authors understand as to "why".

Response: We agree with the reviewer that this scoping paragraph is not very clear. The main framing for the scoping paragraph is that there have been reviews that focus on specific methods to investigate pre-, post-event changes (Moreira et al. 2021) or investigate what subdimensions of social vulnerability are outcomes and pre-conditions for multi-hazard events (Drakes & Tate, 2022), but there is a lack of a general overview of the methods used to investigate vulnerability dynamics. This paper aims to showcase and discuss how different conventional vulnerability assessment methods can be used/tailored to assess dynamics in vulnerability. Per Method, we investigate what data (types) are used, which (sub-)dimensions of vulnerability are considered, what temporal coverage is possible (resolution and time span), what changes in vulnerability can be reported, to which causes/situations these methods are applied and what limitations are reported regarding the method.

2) It would be beneficial if the authors could explain as to why we need to understand/assess V dynamics from the perspective of multi, cascading and aggravating hazards. It is somewhat

articulated that vulnerabilities can be "interacting", btu more tangible substantiating examples would be beneficial. Also, a lot of vulnerability indicators, drivers or dimensions are the same for several hazards (e.g., typical indicators such as age, income, housing type, education level are relevant to consider for floods as well as for heat-related events, storms and others), so why do we need to account for them separately in e.g. cases of consecutive events?

Response: We added a paragraph in Line 58 (section Applied concepts and scope for the analysis) which builds on the short introduction of the concept of vulnerability in the introduction and offer more substance on the pluralistic understanding of vulnerability to introduce and justify the terminology and focus we apply in this study. As part of that, we also will add more information on common dimensions of vulnerability and introduce the sub-dimensions as recently proposed by Stolte et al. 2024. Similarly, we add a paragraph to do the same for dynamic vulnerability which will be accompanied by a visualisation to illustrate different vulnerability dynamics.

3) Is the overall rationale that vulnerability is also driven by impact, so vulnerability is dynamic as a result of a hazard in addition to its own inherent dynamics? Isn't this what is called dynamic risk?

Response: We would not claim that the overall rationale is that vulnerability is driven by impacts alone; there are more causes that influence the vulnerability. Underlying socioeconomic developments, for example (e.g., how residents use left-over money to either invest in DRM or buy beautiful property at the seashore at risk of seasonal flooding), have effects on vulnerability as well. Similarly, interactions between hazards and respective hazard-related impact drivers can also influence the vulnerability (e.g., a person might be able to withstand a certain strong wind when walking on the sidewalk, or they could wade through knee-deep water, but they might not be able to withstand a combination of the two).

4) Line 55 – if the study points our gaps and provides synthesis – it is not a roadmap. The actual contribution of the paper stops at synthesizing gaps and advances

Response: We agree with the reviewer and have reframed the key outputs of this study following this substantive review process.

5) Main criticism concerns methods- I do not consider methodological choices of the authors justified enough not to adhere to the protocol of systematic review. The research question of the study warrants a systematic review, and the field is homogenous enough to pursue it (as previous reviews have done successfully). The search keywords (flood, vulnerability) position the sample firmly into risk and adaptation literature, thus there is little challenge of dealing with the definition of vulnerability from epidemiology for example. In this vein, the justification of following a semi-systematic review is weak.

Response: In line with the feedback from the other reviewer and in response to other comments by this reviewer (see our response to comment 6) for example), we decided to change the methodological approach and scope for this paper. Instead of doing a (semi-)systematic or comprehensive study, we aim for an showcase of how conventional vulnerability assessment methods (curves, indicators, process-based, statistical, qualitative) can be used/tailored to assess vulnerability dynamics.

We use the search query on Google Scholar (as done in the current version) to identify a starting set of papers that helped refine the key method categories to use, data type groups etc. We then use a

targeted literature review per method to investigate relevant studies that offer more insights into how these methods can be applied to assess vulnerability dynamics.

The approach will not be systematic but a showcase, using similar approaches to influential studies like Ward et al. (2020; https://doi.org/10.1016/j.wasec.2020.100070) or di Angeli et al. (2022; https://doi.org/10.1016/j.ijdrr.2022.102829). We thus believe that the findings can still be relevant ot the community.

6) Google Scholar is not an appropriate search engine as it is guided by algorithms and previous user history.

Response: We thank the reviewer for this crucial comment. It was an honest mistake to base the review on Google Scholar, as we were unaware of this limitation. It's also perhaps relevant to acknowledge that this work started as a spin-off from a larger paper effort, which aims to develop an overview of how vulnerability dynamics are considered in different hazard disciplines (flood hazard research as one of them). We've reflected quite a bit on how to move forward with this paper under review, as the applied search terms as used in this review would return no hits when applied to Scopus. Still, we agree with the critical feedback from the reviewer (see our previous response under 5)).

7) Furthermore, the authors have a very limited search sequence. For example, search for risk assessments could yield more suitable papers, as these which often contain the assessment of vulnerability. Additionally, vulnerability dynamics may be an established term in a very niche theme of multi-hazard research, but it is not a well-established term overall, and there are plenty of studies that do relevant things but do not call it vulnerability dynamics. Overall, the search sequence limits the sample in many ways.

Response: We are convinced that an analysis (refined in line with the other comments from the reviewers) of the methods we have identified in this search can offer relevant insights for researchers who want to further advance the research on vulnerability dynamics. We have doubts that completely different patterns would emerge from this additional/different set of publications. We believe that there are always blind spots and limits to any literature review. As a result, we are reluctant to disregard the effort the co-author team put into reviewing 900+ papers and start from scratch with an alternative set of search terms. We preliminarily tested a search query on Scopus that returns a manageable amount of papers for a review: TITLE-ABS-KEY ( {vulnerability NEAR/3 socio?economic} OR {vulnerability NEAR/3 social} OR {vulnerability NEAR/3 physical} OR {vulnerability NEAR/3 assessment} OR (vulnerability NEAR/3 analysis) OR {coping capacity} OR {adaptive capacity} OR {preparedness} )
 AND TITLE-ABS-KEY ( flood* OR inundation)  AND TITLE-ABS-KEY ( dynamic* OR compound* OR cascading OR {multi?hazard} OR {paired event?} OR {temporal change*} OR {spatio?temporal}). The returns show a similar pattern representing the pluralistic understanding of vulnerability.

8) The authors have a limiting set of inclusion criteria. For example, criteria 5 – the study has to adopt a definition of vulnerability from AR6 is too restricting, because IPCC SREX and AR5 have pretty much the same definition and conceptualization. Even if the conceptualizations in the studies are somewhat different stemming from the evolution of vulnerability concept (in IPCC AR4, 5 and 6) – it is still a doable task to appraise the literature and categorize according to the definitions adopted in this study (AR6).

Response: In line with our response to previous reviewer feedback (comment 2), we clarified the inclusion criteria as well. Amongst which is a wider inclusion of vulnerability definitions. This is in line with the practical review we did, as many studies used very different framings (e.g., talking about adaptive capacity and sensitivity) but were considered in this study. We hope that the clarified discussion of vulnerability and how we apply it in this study has improved this.

9) Lines 90-95 discuss why previous IPCC AR definitions have been excluded. I will challenge this rationale – previous studies can indeed be comparable, where vulnerability as per AR5/6 corresponds to adaptive capacity + sensitivity. Previous reviews have done it that way, and there is plenty of studies that point out the evolution of risk and vulnerability frameworks where AR4 vulnerability is comparable to AR5/6 risk, and AR576 vulnerability is adaptive capacity + sensitivity in later frameworks.

Response: We agree, as elaborated upon in our response to the previous comment.

10) The study has a very extensive section on limitations, mainly justifying their methodological choices and treating many of these as inherent review choices. The comments above highlight that many if these limitations could have been overcome and some choices are not justified enough.

Response: We hope that with the revised methodological approach and refined analysis, we were able to address some of the discussed limitations and offer valuable findings to the research community.

---

## Author Comment (AC3)

**Reviewer 1**

Specific comments and technical corrections of "Stocktaking of methods for assessing dynamic vulnerability in the context of flood hazard research"

**Overview**

This article aims to systematically review methods used to assess dynamic vulnerability in the context of floods. The authors also state that they compiled their findings about the drivers and effects of vulnerability dynamics in a dataset (but do not make this available to the reviewer as far as I can tell). The concept of dynamic vulnerability is important for many use cases in both research and practice. Unfortunately, the manuscript is unorganized, often vague, and many of the claims are imprecise. There are signs in the writing that the authors did not finish revising and editing the paper (e.g., an incomplete sentence L180-181 among other sloppy instances). These are limitations that a careful revision could overcome if the editor was forgiving. However, the review is not systematic and that is the main potential value of the manuscript. To overcome this challenge, the review needs a more rigorous sampling strategy, more distinctive and clear conceptual classifications, and an insights-driven synthesis approach.

Response: We want to thank the reviewer for the extensive review provided. We appreciate the time this reviewer took to offer their reflection on the paper, and we intend to use this review to sharpen the paper's scope and address concerns about its scientific relevance. As we understand, the main critiques relate to a mismatch between the planned scope of the article and the methodological choices that were made, along with the reviewer's confusion about the objective of the article stemming from an insufficient elaboration on said scope and the key terminology around vulnerability and dynamics. We understand the reviewer's concerns and have reflected on several options that we hope help to address them.

We acknowledge the ambiguity in the introduced concepts and categories used for the analysis. We intend to better contextualize this research by defining the term of vulnerability and articulating how we are using it in our review, along with the term 'vulnerability dynamics'. For the latter, we don't want to use earlier categorizations (e.g., based on de Ruiter & van Loon, 2023) as a starting point but as a subject for investigation and thus a potential outcome.

Second, we acknowledge that despite our initial claim, the research conducted cannot be called systematic. We discussed options to redo our search by including Scopus and/or Web of Science. However, the original search terms do not return sufficient substance on either search platform, and would necessitate an entirely different strategy for querying these databases and thus require starting from scratch. While we acknowledge that the paper requires significant revisions, we do not believe that re-structuring our entire literature search is necessary to address the reviewer's comments. Instead, we propose that we would more clearly articulate that the purpose of this manuscript is to showcase methods that have been or could be used to assess dynamic vulnerability in the context of flood-related hazards and their impacts. More specifically, we investigate how conventional vulnerability assessment methods (e.g., social vulnerability indices, physics- and process-based models, and statistical and narrative-based methods) can be used or tailored to assess vulnerability dynamics. Instead of using a set of vulnerability categories (as done in the original version), we intend to reorganize the existing material to focus on the methods themselves. We specifically investigate what data are used, which (sub-)dimensions of vulnerability are considered, how these methods capture temporal dimensions of vulnerability dynamics (time span and resolution), or not, and what changes were investigated using these methods, if any. In

restructuring the manuscript to achieve this goal, we can leverage all of the papers we have already collected and reviewed as part of our initial submission.

Third, we intend to discuss whether there are any patterns visible regarding specific causes or situations that result in vulnerability dynamics. This should offer some evidence to reflect on possible ways to categorize vulnerability dynamics, also in the light of already proposed conventions (e.g. de Ruiter & Van Loon, 2023). We do not want to create one based on the evidence we collect, but hope it could be a starting point for future attempts to come up with a relevant categorization of different vulnerability dynamics. Lastly, we also want to articulate limitations of the methods introduced in the papers that we included in our review.

We believe such an analysis of the existing literature still has significance, even if it is not structured as a systematic review. As the reviewer noted, vulnerability research is vast and (partly) ambiguous. Vulnerability dynamics are a subject of interest at the moment, but there are limited insights into how to best assess vulnerability dynamics and their contribution to risk profiles or outcomes. With our review, we want to showcase the existing research in this area, focusing on what has been used/produced in relation to the flood (disaster) risk management community.

We are convinced that the data we have collected offers a promising starting point for providing relevant insights to the community. However, acknowledging the methodological limitations of our current approach, we apply a targeted literature review to identify relevant studies across different methodological frameworks. We intend to use a similar approach to what has been done by Ward et al. (2020; https://doi.org/10.1016/j.wasec.2020.100070) and di Angeli et al. (2022; https://doi.org/10.1016/j.ijdrr.2022.102829), which are just two of many examples of influential research using targeted (non-systematic) literature reviews. With these changes to our analysis and intended scope , we offer a starting point for further investigations from within the flood risk management community, which can be linked to investigations from other hazard communities (e.g., fire, drought) in the future.

Attached, we include line-by-line responses to the reviewer's comments.

Section Comments

Abstract

1. Difficult to parse because of vague language choices and a lack of examples. For example, the authors don't define "dynamic vulnerability." They explain that dynamic vulnerability is driven by a wide range of characteristics and that it is critical to understand flood risk.

Response: We refine the abstract (including the updated scope and narrative for the paper), and included better introductions of key concepts.

2. The abstract should define key concepts, such as dynamic vulnerability. Perhapsthe authors feel that the term is self-evident or widely understood, but it is ambiguous to me and their sample of 28 studies suggests it is not a widespread concept. The second paragraph of the Intro suggests that the authors recognize vulnerability is a pluralistic concept and needs a definition.

Response: As mentioned in the previous response, we will significantly rewrite the abstract, introducing the key term of dynamic vulnerability.

3. The authors should consider whether their four categories of vulnerability dynamics are concepts that can be grouped together (the events categories probably can be, but

"underlying dynamics" seems like a different conceptual entity) and whether it is appropriate to call them "vulnerability dynamics."

Response: As discussed in more detail in our responses regarding the method section, we've scrapped these four categories as a starting point and rather open up the discussions on what dynamics are discussed/subject to in the reviewed studies as part of the results.

4. It is difficult to know what this study is about by reading the abstract.

Response: We hope the abstract's refined version is now clearer. This paper aims to showcase and discuss how different conventional vulnerability assessment methods can be used/tailored to assess dynamics in vulnerability. Per Method, we investigate what data (types) are used, which (sub-)dimensions of vulnerability are considered, what temporal coverage is possible (resolution and time span), what changes in vulnerability can be reported, to which causes/situations these methods are applied and what limitations are reported regarding the method.

5. It seems unfair to highlight in the abstract that a key methodological gap is the limited integration for multi-hazard. The title and beginning of the abstract explicitly frames the study "in the context of flood hazard research."

Response: We have rewritten the abstract significantly to offer a more nuanced reflection on the observed patterns and limitations of methods for dynamic vulnerability assessment.

Introduction

1. Imprecise writing. A few early examples:
   1.1. What do the authors mean by "robust protective infrastructure?"

Response: We went through the introduction and improved the writing. As such, we removed some of the more vague sentences or offered more elaboration (e.g. explaining the concept of dynamic vulnerability (l.33-35).

   1.2. The first paragraph of the introduction talks about the number of flood events reported in a database. The following sentences are about "this increase." Do the authors really want to focus on reported events in a database? It seems like with the reference to vulnerability (also the central framing of the paper) they mean to talk about an increase in flood impacts over time?

Response: The citation refers not to the number of flood events, but the number of disaster events related to floods. With this, we want to show that the number of floods with catastrophic consequences is increasing according to existing databases. We refined the opening paragraph as suggested by the reviewer. We now use it to link flood impacts over time to the components of risk which will be defined in that paragraph as well. The paragraph thus reads as follows:

   "*To be added*"

   1.3. L41 – "These challenges are evident in assessments of…" -> how? It would be helpful to have an illustration of how the challenges are evident instead of just saying they are.

*Response: We replace the sentence starting in line 41 with an example from Moreira et al. 2021 and de Ruiter and van Loon, 2022) and further broaden up the relevance of dynamic vulnerability in the context of multi-hazard research, which thus reads as follows:*

> "*To be added*"

1.4.    L46 – I don't know what the authors mean by the "what" of assessments or the "why" and "how." Do they mean vulnerability in "what," "how" and "why?" Please be more specific. The following examples do not clarify the what/why/how distinctions. I'm not sure how to place the subsequent examples in the context of the preceding sentences. This is the scoping paragraph of the paper. If the reader does not understand these points, they will struggle to understand the purpose or contributions of this paper. The title and abstract make the review seem focused on methods of accounting for or measuring dynamic vulnerability, but this scoping paragraph seems disconnected from that framing.

*Response: We agree with the reviewer that this scoping paragraph is not very clear. The main framing for the scoping paragraph is that there have been reviews that focus on specific methods to investigate pre-, post-event changes (Moreira et al. 2021) or investigate what subdimensions of social vulnerability are outcomes and pre-conditions for multi-hazard events (Drakes & Tate, 2022), but there is a lack of a general overview of the methods used to investigate dynamics of different vulnerability components. With this research, we want to address this gap by investigating which methods have been applied to assess vulnerability dynamics, and which data and dimensions of vulnerability are commonly used. We also explain that part of the scope is to collect the findings about vulnerability changes and potential cause-and-effect relations. We have rewritten the paragraph as follows:*

> "*To be added*"

1.5.    The authors cite Simpson et al., 2021 but make no mention of their proposed "response" dimension that has an implicit dynamic component to it. How can the authors reconcile the "response" dimension to complex risk assessment in that study with their "dynamic vulnerability" framing here? I'm sure that is possible to do, but it is not done here. I think it is necessary to do because these definitions are fickle and the article presents itself as a review, so should aim to be comprehensive and to clarify muddled, overlapping, and imprecise concepts in the field.

*Response: As mentioned in a previous comment (Introduction 1.1), we refine the introduction of the concept for dynamic vulnerability and, as part of that, touch more explicitly on the relevance of recovery and other decisions as part of the DRM cycle as drivers of changes in vulnerability.*

1.6.    L52 – "a comprehensive overview of approaches for assessing dynamic flood vulnerability" in reviewing only 28 studies? This is not a good sign. I'm worried the authors are too attached to this "dynamic vulnerability" language choice instead of studying the underlying phenomena and processes required to develop a rich sample and understanding. Many fields like economics and sociology study things that can fit into "vulnerability" but may not use that language choice. One reason is that, as the authors indicate, vulnerability is multifaceted and pluralistic so some fields like to be more specific about what they are studying than using a vague

catch-all term. Another reason is that there is increasing attention in some fields to avoiding stigmatizing language such as "vulnerability" - https://www.nih.gov/about-nih/what-we-do/science-health-publictrust/perspectives/writing-respectfully-person-first-identity-first-language.

Response: We did not review 28 papers, but more than 950 to arrive at a small set of relevant studies. As we will discuss in more detail in the comments (Methods 7.) regarding the methodological choices, we change the framing of this study to be 1) not systematic, 2) offer a perspective limited by a narrowed terminological focus. It reads as follows:

"*To be added*"

Methods

1.  Section 2.1: I'm not convinced these are "types" of "categories" of dynamic vulnerability. A "type" or "category" of vulnerability might be "financial vulnerability." The choice to call these concepts "categories of dynamic vulnerability" is a major concern for me. I understand that the authors base this on a previous paper, but that does not mean it is a useful conceptual device for classifying approaches. Figure 1 makes panels 1-4 seem like the response of vulnerability to different exogenous events (sometimes flood-related, sometimes not). Not sure that characterizing the sequence of events which the icons/colors represent) is a "category" of dynamic vulnerability because there can be so much heterogeneity in the vulnerability to response to any event or sequence of events. It's such a limited representation of the dynamics surrounding agent decisionmaking that it's distracting. For example, if a person loses their job, they are most likely more susceptible to some forms of harm from flooding than they were before. If a person gets a raise, they are potentially less susceptible to some forms of harm from flooding than they were before. For instance, they could directly cope with economic flood damage or invest in protective infrastructure. But they could also increase their spending and take on more credit, possibly having a lower budget or financial capacity to deal with harms associated with flooding. In summary, I think "vulnerability dynamics" can refer to the change in vulnerability surrounding any of the individual exogenous events or an entire span of events. I don't see how the number and type of events are different categories of vulnerability dynamics. They seem more like different realizations of exogenous events that different people/organizations/institutions may respond differently to.

Response: We thank the reviewer for this reflection. We agree and had similar discussions about the usefulness of these types/categories and whether they should be a frame for the analysis or an outcome from the analysis. Based on the feedback from the reviewer, we reopened the discussion and decided to remove Figure 1 (and replace it with a more illustrative one outlining different vulnerability dynamics). We use information now captured by the figure for a more extensive introduction of the concept of vulnerability and dynamic vulnerability. We replaced lines 59 to 79 with the following. It reads as follows:

"*To be added*"

2.  More on Figure 1 - I think it is attractive and promising, but the icons on the top are not self-evident, there is no link to the definitions of the categories and the 4 panels (is it supposed to be 1:1?), I'm not sure what is the purpose of the magnifying glasses, and the lack of a legend makes aspects uninterpretable. I think part of the problem is how abstract the concepts of vulnerability and dynamic vulnerability still are at this point of the manuscript.

The Figure 1 might work better if it sticks to a specific example, like a structure's susceptibility to damage for a fixed level of flooding, over different sequences of shocks. It might work to have a few rows with different examples, demonstrating how vulnerability dynamics look different for different types of actors and different sequences of events/shocks.

Response: As in our previous comment (Methods 1.), we decided to remove the figure and replace it with a more illustrative visualization of vulnerability dynamics.

3.   The concept of vulnerability is so abstract here and it would be very helpful to have more concrete examples. Otherwise, it is ambiguous and distracting to keep seeing references like "financial vulnerability." What does this mean? I think the reader needs guidance on a specific example of a vulnerability and how it can change over time. On the financial vulnerability reference, I have a lot of confusion. Referring to Thomson et al. 2023, they write "[Our] approach utilizes a series of … to improve understanding of how systemic financial risk could arise from flood impacts to residential properties. As such, this work illustrates a more nuanced approach to evaluating flood-induced financial vulnerabilities." I know this is not the paper under review, but the present study characterizes Thomson as assessing "financial vulnerability" so I wanted to raise the relevant context. Is systemic financial risk a type of flood-induced financial vulnerability? Perhaps, but it's not self-evident and needs much more development and spoon-feeding as a concept. What is the element at risk? What are the specific social, economic, and physical characteristics of the element at risk that characterize "financial vulnerability?" Why isn't "financial vulnerability" a vulnerability category? Are we just talking about income?

Response: We added a paragraph in Line 58 (section Applied concepts and scope for the analysis) which builds on the short introduction of the concept of vulnerability in the introduction and offer more substance on the pluralistic understanding of vulnerability to introduce and justify the terminology and focus we apply in this study (see response to comment Method 1). As part of that, we will also add more information on common dimensions of vulnerability and introduce the sub-dimensions recently proposed by Stolte et al. (2024). Similarly, we add a paragraph to do the same for dynamic vulnerability which will be accompanied by a visualisation to illustrate different vulnerability dynamics.

4.   I don't see the four categories of dynamic vulnerability as a useful division of concepts. The examples for co-occurring event dynamics and underlying dynamics categories seem very similar. For example, a pandemic lockdown can induce an economic crisis. I think the authors need a better classification scheme or much more justification for the one proposed here. On L59 they say "we identify four categories of dynamic vulnerability essential for flood risk assessment (Figure 1)" but don't say how they identified those categories.

Response: We've addressed this comment in response to a previous reviewer comment (Methods 1.).

5.   The authors claim the review is systematic (L81) but it's not clear how to determine what a successful systematic review "of dynamic flood vulnerability" looks like. Is it models of dynamic flood vulnerability? Is it drivers? Is it data sources for evaluating? The authors may feel that my subsequent comments miss the mark because I focus on things outside their scope, but the issue is that they don't define their scope clearly in the Introduction. "Approaches for assessing dynamic flood vulnerability" is general and vague, yet only 28

studies in the sample? And I see that none of the studies that immediately come to my mind (anchored in the Thomson example thinking about "financial vulnerability" which I think is really just income) are cited in this paper, like:

    5.1.    Gallagher, Justin, and Daniel Hartley. 2017. "Household Finance after a Natural Disaster: The Case of Hurricane Katrina." American Economic Journal: Economic Policy 9 (3): 199–228.

    5.2.    Deryugina, Tatyana, Laura Kawano, and Steven Levitt. 2018. "The Economic Impact of Hurricane Katrina on Its Victims: Evidence from Individual Tax Returns." American Economic Journal: Applied Economics 10 (2): 202–33.

    5.3.    Deryugina, Tatyana. 2017. "The Fiscal Cost of Hurricanes: Disaster Aid versus Social Insurance." American Economic Journal: Economic Policy 9 (3): 168–98.

    5.4.    Or even Kousky, C., Palim, M., & Pan, Y. (2020). Flood damage and mortgage credit risk: A case study of Hurricane Harvey. Journal of Housing Research, 29(sup1), S86-S120 which is a motivating reference for Thomson et al. (2023).

Response: We will address the critique regarding the systematic aspect in the following comment (Method 7.) and have addressed the comment regarding the unclear scope as part of a previous response (Introduction 1.4). We thank the reviewer for offering these additional papers and will decide whether to include those (along with others as explained in our response in Method 6.).

6.    Also on the systematic claim, I don't understand how investments in flood-risk reduction are out of scope of a review focused on dynamic vulnerability. These investments often occur in the context of post-disaster intergovernmental transfers and disaster aid. For example, see

    6.1.    Deryugina, Tatyana. 2017. "The Fiscal Cost of Hurricanes: Disaster Aid versus Social Insurance." American Economic Journal: Economic Policy 9 (3): 168–98 and

    6.2.    Davlasheridze, Meri, Karen Fisher-Vanden, and H. Allen Klaiber. "The effects of adaptation measures on hurricane induced property losses: Which FEMA investments have the highest returns?." Journal of Environmental Economics and Management 81 (2017): 93- 114. This is also a particularly interesting dynamic because of connections to gentrification. Areas that become safer may become more desirable to live, attracting investment and increasing overall exposure (and potentially risk!) while also potentially displacing people with lower incomes.

Response: In line with our response to the previous comment, we consider these additional papers.

7.    I am surprised to see that Google Scholar was used for obtaining an initial sample. A recent article in PlosOne recommends in its abstract that "whilst Google Scholar can find much grey literature and specific, known studies, it should not be used alone for systematic review searches. Rather, it forms a powerful addition to other traditional search methods." See Haddaway, Neal Robert, et al. "The role of Google Scholar in evidence reviews and its applicability to grey literature searching." PloS one 10.9 (2015): e0138237. One of the most important reasons not to use Google Scholar is that it is much more difficult to reproduce and replicate a search than using a more typical database for a systematic review such as Scopus or Web of Science. The large number of not peer-reviewed studies in their initial sample and studies excluded based on title and abstract make the use of this search engine questionable as well.

Response: We thank the reviewer for this crucial comment. It was an honest mistake to base the review on Google Scholar, as we were unaware of this limitation. It's also perhaps relevant to acknowledge that this work started as a spin-off from a larger paper effort, which aims to develop an

overview of how vulnerability dynamics are considered in different hazard disciplines (flood hazard research as one of them). We've reflected quite a bit on how to move forward with this paper under review, as the applied search terms used in this review would return no hits when applied to Scopus, but we agree with the critical feedback from the reviewer.

We are convinced that an analysis (refined in line with the other comments from the reviewers) of the methods we have identified in this search can offer relevant insights for researchers who want to further advance the research on vulnerability dynamics. We have doubts that completely different patterns would emerge from this additional/different set of publications. We believe there are always blind spots and limits to any literature review. As a result, we are reluctant to disregard the effort the co-author team put into reviewing 900+ papers and start from scratch with an alternative set of search terms. We preliminarily tested a search query on Scopus that returns a manageable amount of papers for a review: TITLE-ABS-KEY ( {vulnerability NEAR/3 socio?economic}  OR {vulnerability NEAR/3 social} OR {vulnerability NEAR/3 physical} OR {vulnerability NEAR/3 assessment} OR (vulnerability NEAR/3 analysis) OR {coping capacity} OR {adaptive capacity} OR {preparedness} )
 AND TITLE-ABS-KEY ( flood* OR inundation)  AND TITLE-ABS-KEY ( dynamic* OR compound* OR cascading OR {multi?hazard} OR {paired event?} OR {temporal change*} OR {spatio?temporal}). The returns seem to show a similar pattern representing the pluralistic understanding of vulnerability.

If we want to keep the analysis of the current paper, the review won't be systematic. For example, if we were to use an alternative set of broader search terms that builds on what we learned from the writing process and the reviews, we would have to argue why we use different search queries on different platforms. The more straightforward solution seems to be to reframe this study by eliminating claims about 'comprehensiveness' or 'systematic review'. We believe that we can still make a case for the relevance of such a study as a showcase of methods, especially given the diverse types of terminology and foci on vulnerability dimensions. Accordingly, we proposed to refine the scope as outlined in our response to a previous reviewer comment (Introduction 1.4).

As a result, we decided to change the methodological approach of this paper: We use the search query on Google Scholar (as done in the current version) to identify a starting set of papers that helped refine the key method categories to use, data type groups etc. We then use a targeted literature review per method to investigate relevant studies that offer more insights into how these methods can be applied to assess vulnerability dynamics.

The approach will not be systematic but a showcase, using similar approaches to influential studies like Ward et al. (2020; https://doi.org/10.1016/j.wasec.2020.100070) or di Angeli et al. (2022; https://doi.org/10.1016/j.ijdrr.2022.102829). We thus believe that the findings can still be relevant ot the community.

8. In addition to the use of Google Scholar, I find several aspects of the article sample process flawed:

    8.1. Criterion iii requires that the article be freely accessible to the reviewers. Was interlibrary loan not available to the authors? This can unnecessarily limit the sample size and with authors from so many institutions it is surprising that this is a limiting factor for including some studies. Dropping 35 studies for this reason seems

Response: In Figure 2, we lumped together papers that were not accessible due to criteria i and iii. Due to criterion i, nine publications were excluded. We revisited the list of papers to check which could be made accessible through the different institutions of the co-authors.

8.2.	Criterion v requires that the study adopt a definition of vulnerability consistent with the IPCC or UNDRR. This seems to substantially limit the authors' ability to systematically review dynamic vulnerability and how it is studied in research. It seems it would be more effective to search for research that focuses on specific drivers of dynamic vulnerability (like recovery, changes in income, adaptation funding, infrastructure projects, etc.,). This criterion seems to induce a selection bias for disciplines that utilize vocabulary preferred by the authors. I think the review would have more useful insights if it expanded its search to include studies that investigate

Response: As outlined in our comment to earlier reviewer feedback, we offer a more extensive reflection on the definitions of vulnerability, and how we apply those in this study (Methods 3. and Introduction 1.3). As such, we will revise criterion v.

8.3.	Criterion vi requires that the study address one of the vulnerability dynamics identified in Figure 1. Seeing that these dynamics were a priori determined by the authors, instead of informed by a review of potentially relevant literature, is concerning and demands that the authors more transparently define how they identified these categories. They do not do this in the beginning of section 2.1.

Response: In line with our comment on an earlier feedback (Method 1.), we remove this criterion. For clarification, it was not really applied as an exclusion criterion but rather to categorize different vulnerability dynamics.

8.4.	Criterion vii requires that the study provide details allowing replication. This is a major limiting factor. What degree of published studies adequately provide details allowing replication? Few studies in the area of flood risk assessment even make their data available, a requirement for replication.

Response: We clarified criterion vii by removing the condition that it is replicable. We did not go into depth in the evaluation whether the method is replicable, but rather aimed with this criterion at making sure that the considered publication offers insights into data, methods, and algorithms, and not just draws the result out of a black-box.

8.5.	Criterion viii requires that the study is a case study. Why? What is an example of a study that is not a case study and was excluded for that reason?

Response: Actually, most publications we found were case studies. We wanted to exclude reviews or perspective papers that offer a theoretical reflection without practical use. We've clarified this criterion to capture this intention.

9.	All of the criteria for screening need a justification, and many of them lack one.

Response: We updated lines 87ff to refine some of the inclusion criteria and provide justification for some of them. It reads as follows:

	"*To be added*"

10.	How many of the studies in the final sample of 28 come from the additional papers added by collaborators?

Response: In total, 12 out of the 28 papers come from the papers added by collaborators (including the paper from Phifer et al. 1988).

11.   On Line 98 they say they categorize methods into five groups but then proceed to only define four groups.

Response: We corrected this mistake

Results

1.   I am skeptical about their Google Scholar search. The first study mentioned in the results section is Phifer et al., 1988, which must be added by one of the authors because it has no mention of "vulnerability assessment" or "vulnerability analysis" and these are requirements for search terms. If the first relevant study does not have these search terms, why do the authors think these are reliable search terms to guide a "systematic" search? The authors would likely better serve their review goal if they started from their known examples of studies and worked backwards to identify search terms.

Response: As outlined in our comment to the previous reviewer feedback (Method 7.), we want to re-frame this study as an showcase and not a systematic review.

2.   The Phifer et al., 1988 article raises another issue – is this article about vulnerability or impacts? The title is "The Impact [emphasis mine] of Natural Disaster on the Health of Older Adults: A Multiwave Prospective Study" and in the abstract states "The present study examined the impact [emphasis mine] of flood exposure on the physical health of this sample..." Revising the IPCC definition, "vulnerability refers to the social, economic, and physical characteristics of an element at risk that make it susceptible to harm in the event of exposure to a hazard." I suppose the element at risk here is the health of older adults. In that case, how is the study's focus not about the element at risk of health of older adults? One could argue that declining health can make one more susceptible to future declines in health. This raises the concern that the authors of the review are not clear about vulnerability "in what" and the review can complicate the concept of vulnerability in the reader's mind instead of clarifying and enriching it.

Response: In this specific example, we would argue that the element at risk is elderly/humans. Their predisposition to adverse effects is driven by their health (one of the vulnerability sub-dimensions of social vulnerability). We take the comment about the unclear definition of vulnerability seriously and have extended the elaboration on the concept of vulnerability and how it is applied in this study, as mentioned in previous comments (Method 1., Method 3.)

3.   The relatively high proportion of studies in Europe raises concerns about the sample of studies as well, given the high representation of European authors. Given the content and focus of the Phifer et al., 1988 article, I can think of a number of health and economic studies that are appropriate for the authors to include (such as those I cited earlier). I think the authors would better serve their "systematic" review goals if they used a snowball sampling approach from the studies they knew, looked at citations/references for relevant studies to include, and then worked backwards from this sample to identify appropriate screening terms for a broader review of relevant articles.

Response: We thank the reviewer for this suggestion. We've considered this approach to extend and refine this research, but decided to go for a  targeted literature review instead.

4.   The idea that only two studies assess vulnerability dynamics due to single flood events is wrong. I know this because one of the studies they describe in the single event section investigates the impact of the 2015 Mozambique flood on household consumption and

poverty levels, using a difference-in-difference approach "to quantify changes in vulnerability attributable to the flood event." If income, poverty, welfare, etc., are all forms of vulnerability (this was not clear to me as there was no description before on how one can measure vulnerability), then this review is missing a sizable number of studies in economics that use a wide variety of methods in causal inference to evaluate the impact of floods on households. In addition to the studies I cited before, there are a number of studies that evaluate how disasters and other exogenous events (like risk communication efforts) affect flood insurance take-up and flood-risk reduction investments. The review misses key studies from authors such as Kunreuther, Kousky, Czajkowski, Botzen, Bubeck, and Gallagher (and these are authors I know without dynamic vulnerability being a specific area of focus for me – surely they miss many more key studies and authors). For example:

4.1.  Bubeck, P., Berghäuser, L., Hudson, P., & Thieken, A. H. (2020). Using panel data to understand the dynamics of human behavior in response to flooding. Risk Analysis, 40(11), 2340-2359.

4.2.  Bubeck, P., Botzen, W. J., Kreibich, H., & Aerts, J. C. (2013). Detailed insights into the influence of flood-coping appraisals on mitigation behaviour. Global environmental change, 23(5), 1327-1338.

4.3.  Botzen, W. W., Kunreuther, H., Czajkowski, J., & de Moel, H. (2019). Adoption of individual flood damage mitigation measures in New York City: An extension of protection motivation theory. Risk analysis, 39(10), 2143-2159.

4.4.  Gallagher, J. (2014). Learning about an infrequent event: Evidence from flood insurance take-up in the United States. American Economic Journal: Applied Economics, 206-233

4.5.  Kousky, C. (2017). Disasters as learning experiences or disasters as policy opportunities? Examining flood insurance purchases after hurricanes. Risk analysis, 37(3), 517-530

Response: We again thank this reviewer for suggesting additional publications and authors (teams). We collected further publications to consider as part of the refined methodological approach. The definition of vulnerability was clarified as outlined in comment (Method 3.).

5.  The results section describes studies and characterizes their features (e.g., in Figure 4) in ways that do not convey insights. They hardly break down Figure 4 in the results text. Part of the problem is that the review does not state clear research questions so some of the results seem non sequitur. For example, starting on L154, the authors discuss the timing of data collection. They discuss this for 4 sentences. This is in a paragraph about "the time intervals between the consecutive events, the number of flood events considered, and the duration of the analyses." With this paragraph's stated focus, why is it "important to note that Kreibech et al. (2017, 2023) use a literature review considering various reports and publications from different years to assess vulnerability dynamics, while we took the date of the scientific publication to determine the time lag between events and data collection?" This long discussion seems irrelevant. What does any of this have to do with vulnerability dynamics? What does it have to do with methods for assessing vulnerability dynamics?

Response: As part of the refined scope (see response to comment Introduction 1.4), we clarified a structure and order for the results section, which builds more closely around an (updated) Figure 4.

6.  L179: "The studies mentioned above capture the vulnerability dynamics regarding different elements at risk and vulnerability dimensions. While Phifer et al. (1988) focus on human health and well-being, taking into account demographic, economic, and health dimensions

of vulnerability," -> that's the end of the sentence! This is unacceptable for a submission to such an esteemed journal.

Response: We corrected this mistake.

7. L196 – this paragraph's focus on elements at risk and contributing vulnerability dimensions is shallow and vague. It would be much more helpful for the authors to explicitly state what the vulnerability dimensions are and what the impact of floods are on these dimensions. For example, what does the following mean? "Albulescu and Armas (2024) use augmented impact chains to express the effects of hazard impacts and risk mitigation measures on vulnerability without focusing on a certain element at risk"

Response: We added more depth to the focus of this paragraph. It reads as follows:

*"To be added"*

8. Section 3.5 is titled "dataset for vulnerability dynamics" but then the text on 235 starts talking about findings from studies. Very unorganized and confusing. It's nice to see some discussion of the findings from these studies, but it is presented in an unorganized way and is hard to follow. The separation into "categories" of consecutive event dynamics, co-occurring hazard dynamics, single-event dynamics, and underlying vulnerability detracts from the flow of the discussion. This separation forces distinctions that move relevant findings apart from each other. Review articles should synthesize in ways that produce insights, not just produce a list of thematically linked findings and leave it to the reader to connect the dots.

Response: In line with earlier comments about removing the types of dynamic vulnerability as an organizing principle for this paper, we've also restructured this section accordingly and hope the reviewer finds it now more useful. Furthermore, as refined in the updated scope (Introduction 1.4), we clarified what we mean by a database of vulnerability dynamics.

Discussion

1. I don't understand how the indicator-based approaches, which the authors state entirely avoid investigating how vulnerability indicators change because of flood events, can be included in this review on vulnerability dynamics. This again supports the idea that the authors' categories of vulnerability dynamics are not helpful for synthesizing vulnerability dynamics. In fact, their categories are more like hazard dynamics (these are mostly hazard realizations over time). Vulnerability dynamics suggest attention to changes in vulnerability dimensions over time.

Response: We've updated this section in the discussion in line with the refined results, methods and scope of the paper. It reads as follows:

*"To be added"*

2. The discussion on "curve methods" L334 is underwhelming. The discussion does not acknowledge a wide literature on flood damage functions and substantial uncertainty in relating flood and structure characteristics to economic damages. There is also a random seeming reference to "storage tanks" in L341. What exactly do the authors mean that a damage curve is static? I think they mean that most damage curves take generally static

characteristics of a structure as input, such as number of stories, foundation type, and first-floor elevation, and produce percent damage estimates for possible inundation depths relative to first floor. But that's not all damage functions, and even these characteristics can change over time. For example, consider the following study that uses building quality as an input variable, a structure characteristic that may change between flooding at time t1 and time t2 if the damage is large enough: Schröter, K., Kreibich, H., Vogel, K., Riggelsen, C., Scherbaum, F., & Merz, B. (2014). How useful are complex flood damage models? Water Resources Research, 50(4), 3378-3395. This topic is complex and would benefit from more thoughtful consideration of relevant literature.

Response: We thank the reviewer for this critical comment. We've included the suggested paper and further investigated additional papers using a targeted review as outlined in our response to a previous reviewer comment (Methdos 7.). It reads as follows:

"*To be added*"

3. Why "agent-based modeling" on L346 but "Agent-based Model" on L349?

Response: We streamlined spelling and grammar throughout the paper.

4. How can the authors claim that "no peer-reviewed study seems to apply [agentbased modeling] to the assessment of dynamics?" L346. For example, see Taberna, A., Filatova, T., Hadjimichael, A., & Noll, B. (2023). Uncertainty in boundedly rational household adaptation to environmental shocks. Proceedings of the National Academy of Sciences, 121(29), e2215675120. I am sure there are many more relevant agent-based modeling studies.

Response: We intended to say that ABMs have not been returned by our search. However, this statement is now redundant given the changed methodological approach and thus different results. It reads as follows:

"*To be added*"

5. The discussion on causality is welcome, but underwhelming. The authors miss many economic studies that use methods in causal inference to identify the effects of floods and other exogenous events on features the authors defined elsewhere as vulnerability (see my many references to economics papers above). The main challenge is the transferability of these findings to the many analytical contexts in which projections of vulnerability dynamics can help inform risk assessments and decisions. Such a discussion is extremely important and would be a very nice contribution to the literature.

Response: We've significantly reworked the results section, including additional papers, and as a result, further substantiated the discussion on causality. It reads as follows:

"*To be added*"

6. I am confused that the authors highlight that they excluded a study by Bryant et al., 2022 because Bryant et al. frame changes as adjustments to hazard rather than vulnerability dynamics. Making a choice based on language rather than whether the object of study is the same is a poor way to do the kind of synthesis this "systematic review" aims for. Committing to strict language choices for a new idea of "vulnerability dynamics" – which is not widespread or standardized (like the way the IPCC is a helpful standard definition for "vulnerability") – is clearly a limiting factor of this review. There are clearly many studies that evaluate changes in vulnerability over time. Leaving these out because of language

choices – as opposed to connecting relevant fields and literatures to create a richer concept of dynamic vulnerability (if that's even the appropriate catch-all for what the authors want to synthesize) – is a failure to be systematic.

Response: Reflecting on this choice, we must agree with the reviewer. In the revised version of this paper, we clarify how we understand and apply the concept of vulnerability and how we deal with the diversity in language and pluralistic definitions of vulnerability. We thus consider Bryant et al. (2022) as a paper, but keep the reflection on the challenge of language in the discussion section.

7. I appreciate their discussion on limitations, but I disagree with the dismissal of these limitations as a normal part of any review and disagree that their process "ensured that we captured a broad and representative sample of studies relevant to dynamic flood vulnerability assessment." (L414) I think the authors miss too many relevant references to consider the review systematic or helpful. The authors are clearly aware that terms such as "panel" and "longitudinal" would be helpful and don't satisfyingly justify why they stick to jargon in their field of (multi-)risk. The topic of study is important and for publication in an interdisciplinary journal such as this one, it is necessary that the authors take efforts to increase representation of studies outside of their own small community.

Response: We hope that with the refined methodological approach, the reviewer agrees that we capture sufficient relevant publications.